# In situ activation of flexible magnetoelectric membrane enhances bone defect repair

Wenwen Liu [1,12], Han Zhao[1,12], Chenguang Zhang[2,12], Shiqi Xu[3], Fengyi Zhang[4], Ling Wei[5], Fangyu Zhu[1], Ying Chen [6], Yumin Chen[1], Ying Huang[1], Mingming Xu[1], Ying He[1], Boon Chin Heng[7], Jinxing Zhang[8], Yang Shen [9], Xuehui Zhang [10] ✉, Houbing Huang [2] ✉, Lili Chen [11] ✉ & Xuliang Deng [1] ✉

For bone defect repair under co-morbidity conditions, the use of biomaterials that can be non-invasively regulated is highly desirable to avoid further complications and to promote osteogenesis. However, it remains a formidable challenge in clinical applications to achieve efficient osteogenesis with stimuli-responsive materials. Here, we develop polarized $CoFe_2O_4@BaTiO_3$/poly(vinylidene fluoridetrifluoroethylene) [P(VDF-TrFE)] core-shell particle-incorporated composite membranes with high magnetoelectric conversion efficiency for activating bone regeneration. An external magnetic field force conduct on the $CoFe_2O_4$ core can increase charge density on the $BaTiO_3$ shell and strengthens the β-phase transition in the P(VDF-TrFE) matrix. This energy conversion increases the membrane surface potential, which hence activates osteogenesis. Skull defect experiments on male rats showed that repeated magnetic field applications on the membranes enhanced bone defect repair, even when osteogenesis repression is elicited by dexamethasone or lipopolysaccharide-induced inflammation. This study provides a strategy of utilizing stimuli-responsive magnetoelectric membranes to efficiently activate osteogenesis in situ.

Co-morbidity conditions, such as wound infection, diabetes, osteoporosis, etc., often pose many difficulties for bone defect repair in the clinic[1-3]. Implanted biomaterials that mimic bone structure and composition should preferably be removed under such co-morbidity conditions, to prevent further complications[4-6], which could result in failed repair or delayed healing[7,8]. Hence, a stimuli-responsive biomaterial with the capacity to reactivate the bone regeneration in situ is much needed for bone defect repair with unanticipated co-morbidity.

Reconstructing the normal physiological electrical microenvironment at the wound/injury site can effectively promote bone defect repair[9-12]. Compared with biomaterials such as gels, scaffolds, or particles, a charged membrane covering the bone defect can maintain healing space as well as induce osteogenesis and vascularization, thus promising for clinical application[13-15]. Moreover, after complete bone healing, the non-absorbable charged membranes can easily be removed without any residual material[10,16]. In the meantime, reactivation of bone regeneration when encountering co-morbidity conditions requires that the charged implant membrane can be regulated on demand. Therefore, it is imperative to develop a stimuli-responsive rechargeable membrane with high energy conversion efficiency, for effective bone regeneration with unanticipated co-morbidity[17,18].

Magnetoelectric materials exhibit unique capacity for tuning electrical properties with magnetic field modulation[19-21]. Nevertheless, due to their significant toxicity, few of them have ever been utilized in the biological field[22]. Magnetoelectric membranes made of $CoFe_2O_4$ particles embedded within a piezoelectric matrix exhibit good biocompatibility and have suitable electrical properties required for bone regeneration[23,24], but it is challenging to reactivate and accurately

A full list of affiliations appears at the end of the paper. ✉e-mail: zhangxuehui@bjmu.edu.cn; hbhuang@bit.edu.cn; chenlili1030@hust.edu.cn; kqdengxuliang@bjmu.edu.cn

provide an optimal electrical microenvironment in situ for bone defect repair because of their relatively low magnetoelectric coupling. Multiphase-structured magnetoelectric materials are expected to have high interfacial coupling efficiency[22], thus their electrical properties can be more efficiently tuned by an applied external magnetic field conveniently to activate its pro-osteogenic effect.

In this study, we fabricated polarized flexible CoFe$_2$O$_4$@BaTiO$_3$/P(VDF-TrFE) core-shell particle-incorporated composite membranes (CSCM). The core-shell crystal lattice structure of CoFe$_2$O$_4$@BaTiO$_3$ nanoparticles[25-27] can enhance the magnetoelectric coupling of the membrane. We show that the markedly increased BTO shell charge density significantly strengthened the reversible β-phase transition of P(VDF-TrFE), leading to maximal repolarization of CSCM under magnetic field loading. (Fig. 1c). Animal models confirmed that CSCM can reactivate an electrical microenvironment for bone regeneration under inflammatory conditions, or when osteogenesis was repressed (Fig. 1b). Compared with the performance of other non-invasive stimuli-responsive materials such as photothermal materials, sonodynamic materials, etc., CSCM possesses the capacity for repeated activation and exhibited the higher osteogenic efficiency even in co-morbidity condition (Fig. 1a).

## Results
### Synthesis and characterization of CSNP and CSCM
CoFe$_2$O$_4$@BaTiO$_3$ core-shell nanoparticles (CSNP) were constructed via the sol-gel method (Supplementary Fig. 1a). CSNP were embedded within the P(VDF-TrFE) matrix to form composite membranes, which were then polarized under corona poling (Supplementary Fig. 2c). The high-resolution Transmission Electron Microscope (TEM) was used to visualize the core-shell structure and particle morphology of CSNP (Fig. 2a, b). The 150 to 300 nm sized nanoparticles had a tight core (CFO) and shell (BTO) crystal lattice with an epitaxial relationship, which could facilitate mechanical force transmission from the CFO core to BTO shell (Fig. 2a)[28]. Compared with CFO nanoparticle-incorporated composite membranes (CCM), the CSNP were uniformly distributed within the membranes, as can be observed from the surface morphology with Scanning Electron Microscopy (SEM) (Supplementary Fig. 1b, c) and the TEM view of the membrane sections (Fig. 2c). The BTO shell layer served as a barrier to preclude direct contact among magnetic particles, and the electrostatic force on the shell surface also reduced the agglomeration phenomenon.

The composition of CSNP was analyzed by X-ray diffraction (XRD) spectroscopy. In XRD, the ratio of CFO/BTO was found to be approximately 1:3, which was the optimal proportion of core-shell

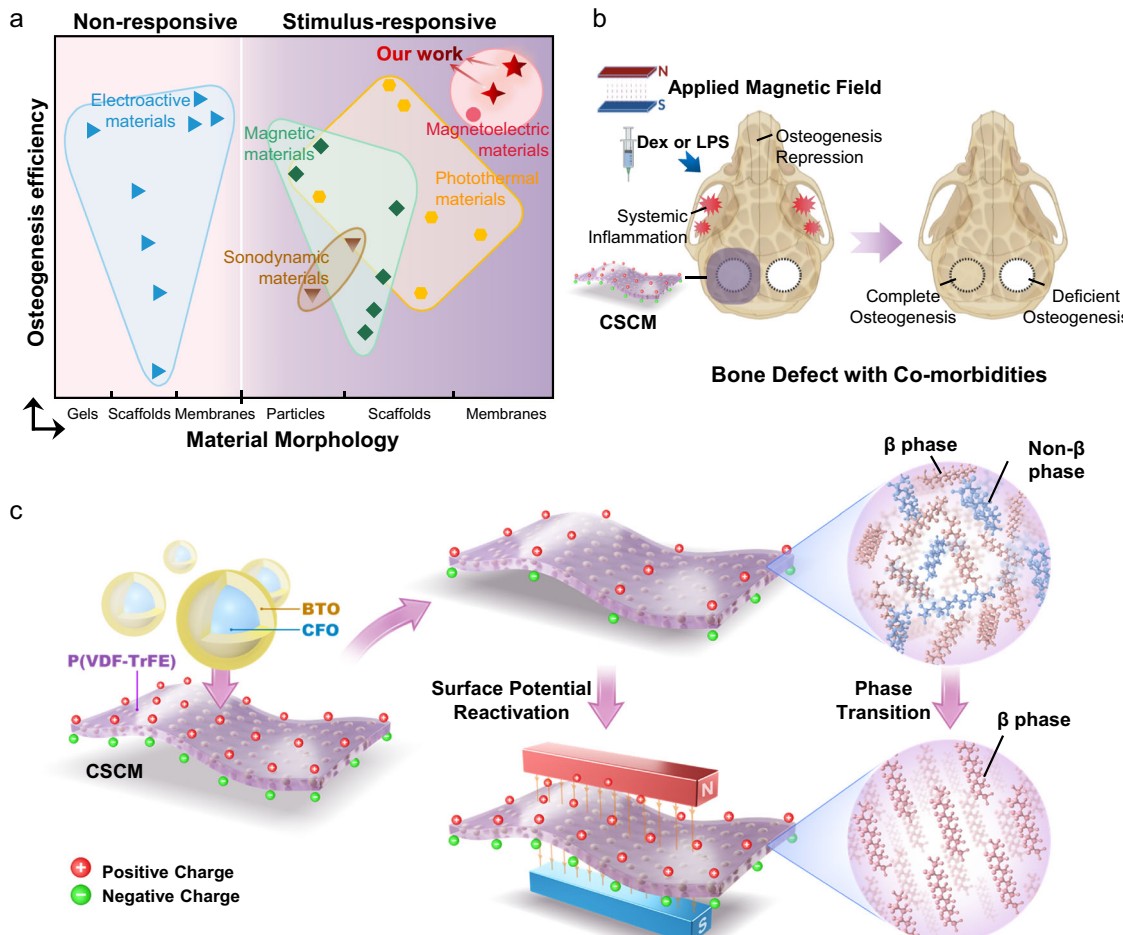

**Fig. 1 | Design and working mechanisms of the CoFe$_2$O$_4$@BaTiO$_3$/P(VDF-TrFE) membrane, which can enhance osteoinductivity on command upon reactivation by a magnetic field. a** Osteogenesis efficiency of CSCM at general state (red asterisk) and under co-morbidity conditions (red four-point star), as compared with electroactive materials (blue triangles), magnetic materials (green diamonds), photothermal materials (yellow hexagons), sonodynamic materials (brown triangles) and magnetoelectric materials (pink dots) under general state. Materials are classified according to the material morphology. The osteogenisis efficiency is represented by the ratio of bone volume to the total volume (BV/TV). Details and values of the aforementioned materials are listed in Table S1. **b** A schematic diagram of magnetic field reactivation of CSCM to repair bone defects under co-morbidity conditions. Created with BioRender.com. **c** A schematic diagram of the core-shell composite membrane (CSCM) reactivated by magnetic field and the internal phase transition.

powder to use in this study (Supplementary Fig. 2d). X-ray photo-electron spectroscopy (XPS) and Fourier transform infrared spectroscopy (FTIR) were used to analyze the chemical bonds and phase structure of CSCM. In the XPS results, C 1 s showed that both CSCM and CCM presented three C states (Supplementary Fig. 2a), corresponding to $-CF_2-$, $-CFH-$, $-CH_2-$, which indicated that the membranes were composed of P(VDF-TrFE) as the matrix[29]. In FTIR (Supplementary Fig. 2b), the three β-phase correlated absorption bands (1288, 850, and 1400 cm$^{-1}$)[30] of CSCM were higher than those of CCM, which indicated higher piezoelectric phase contents in CSCM than in CCM.

The CSCM possessed good flexibility. The tensile strength of the membranes was within the range of 25 - 27 MPa, and the elastic modulus was between 0.4 - 0.6 GPa, which was suitable for clinical application[10]. With increasing CSNP content, the tensile strength and elastic moduli showed slight changes without any statistically significant differences (Supplementary Fig. 3d).

### Activation effects of magnetic field on the surface potential of CSCM

To explore the changes of membraness' surface potential over time, we immersed samples in PBS to simulate in vivo conditions. The membranes were grouped as CSCM-M, CSCM, CCM-M, and CCM. "-M" represents membranes exposed to a direct current (DC) magnetic field. The initial CSCM zeta surface potential was about −70 mV, which was within the range of physiological potential. Both CSCM and CCM displayed a decreasing surface potential trend without magnetic field loading (Supplementary Fig. 4a, b). When the magnetic field was periodically loaded, the surface potential of CSCM was maintained within the physiological potential range. The relatively stable surface potential arose from the dipoles inside CSCM, which were repeatedly reactivated by loading an external magnetic field[31].

To evaluate the reactivation ability of CSCM and the immediate effects of magnetic field loading on the surface potential, we recorded the instantaneous potential change of CSCM when the magnetic field was applied instantly in situ (Supplementary Fig. 6a, b). The results showed that the relative surface potential increased nearly 200 mV with in situ magnetic field loading (Fig. 2f), which indicated that the repowering effect of the magnetic field was direct and instantaneous.

To demonstrate the persistent repowering effect of the magnetic field on the membrane surface potential, we soaked the membranes for different periods and then loaded the magnetic field periodically (Fig. 2g). CSCM displayed higher normalized surface potential than CCM in both the zeta potentiometer and Scanning Kelvin Probe Microscopy (SKPM) measurements. The piezoelectric property of BTO arising from the ferroelectric component contributed to higher surface potential in CSCM[32]. On the other hand, without magnetic field loading, the surface potential of CSCM decreased (Supplementary Fig. 4c, d). Upon repowering by the magnetic field, the decreasing trend of the normalized surface potential was significantly reduced in CSCM (Fig. 2h, i), which proved that the dipoles within CSCM were reactivated by the reloaded magnetic field[33]. To investigate whether surface adsorption of proteins and molecules affects the surface potential, we soaked the material in cell culture medium and detected the zeta surface potential. Both membranes loaded with continuous magnetic field and 12-hour interval magnetic field were able to maintain the surface potential better than non-magnetic field group (Supplementary Fig. 16).

To further investigate the adjustability of membrane surface potential under magnetic field loading, the magnetoelectric coupling coefficient ($\alpha$) of the membranes was determined. The CSCM showed an improved $\alpha$ value of 6.06 mV·cm$^{-1}$·Oe$^{-1}$ at 3000 Oe magnetic field intensity, which was significantly higher than that of CCM (Fig. 2d). This high $\alpha$ endowed CSCM with high surface potential (Fig. 2e) and

extensive tunability. These results thus indicated that CSCM had excellent responsiveness to magnetic stimuli[34].

### Magnetic field drives polarization by β-phase transition

To further explore the underlying mechanisms by which applied magnetic field increases the surface potential of CSCM, the XRD spectroscopy analysis (Supplementary Fig. 5e) was carried out with or without magnetic field loading. Within the XRD patterns, it was observed that the intensity of the β-phase of P(VDF-TrFE) in CSCM was significantly increased when the magnetic field was loaded. By contrast, no β-phase change was found in CCM upon magnetic field loading (Fig. 3b). With a strong ferroelectric performance of CSCM, the all-trans molecular configuration of β-phase, exhibited a higher polarization level under magnetic field, which led to increases in the β-phase diffraction peak[29]. Moreover, in the XRD pattern, the β-phase peak position of the CSCM was shifted after magnetic field loading (Fig. 3c). The abscissa 2$\theta$ corresponding to the β-phase peak value, exhibited a left shift (normalized with the last peak as a reference), which was reflected by decreasing 2$\theta$ value[35]. The main explanation for this phenomenon is that the magnetic force drives ferromagnetic CFO and the stress is then fully transferred to the BTO shell[36]. The ferroelectric shell converts the stress into surface charges, which in turn induces the interfacial effect on the P(VDF-TrFE) matrix[36–38]. The interfacial effect enhances the increase in P(VDF-TrFE) interplanar crystal spacing and induces β-phase formation. In conclusion, the strengthening and shifting of the β-phase indicate that the magnetic force on the ferromagnetic CFO core drives the interfacial conformational stress to the outer ferroelectric BTO shell. The shell fully receives the force transmission within a confined space, which facilitates strain delivery on the BTO shell and generates surface charges[39]. The highly efficient energy conversion is accomplished by increased charge density on the BTO shell, which induces the interfacial effect and further enhances the β-phase transition of matrix P(VDF-TrFE).

To verify that the β-phase transition in CSCM was repeatable with magnetic field loading, we conducted a semi-quantitative assay on the β-phase content at different time points (0, 24, 48 h) after magnetic field loading (Fig. 3d, e). The β-phase content in CSCM was altered according to periodical magnetic field loading, which proved that the β-phase transition in CSCM could be reactivated repeatedly. After removing the magnetic field, the β-phase content in CSCM showed a decrease but was increased again upon magnetic field reloading. The residual polarization ($P_r$) in the P-E loops of CSCM were consistent with the trend of β-phase content in the XRD results (Fig. 3f). To further demonstrate the mechanism that the magnetic field modulates the surface potential by enhancing the β-phase transition, we detected cycles of the β phase changes. Results showed that β-phase transition could be enhanced by applying a magnetic field in a long-term cycle. (Supplementary Fig. 17). The repeated loading of the magnetic field led to repeated activation and repolarization of the membranes, which enabled the adjustment and rechargeability of CSCM.

### Core-shell particle/P(VDF-TrFE) interface polarization for repowering membrane surface potential

SKPM was used to investigate the interfacial polarization between CSNP and the P(VDF-TrFE), together with surface potential reactivation with magnetic field loading (Fig. 3g). The results demonstrated that the interface polarization effect increased the surface potential between the BTO shell and the matrix P(VDF-TrFE). Furthermore, the overall surface potential of the CSCM increased simultaneously when the magnetic field was loaded. We also quantified the surface potential at the interface layer between the BTO shell and the matrix P(VDF-TrFE) (≈0.4 μm), with and without magnetic field loading. The surface potential at the interface layer $\Delta$SP ($\Delta$SP = SP$_{Interface}$ − SP$_{Matrix}$) increased significantly upon applying a magnetic field (Fig. 3i). The increased $\Delta$SP suggested that polarized charge accumulated at the interface layer

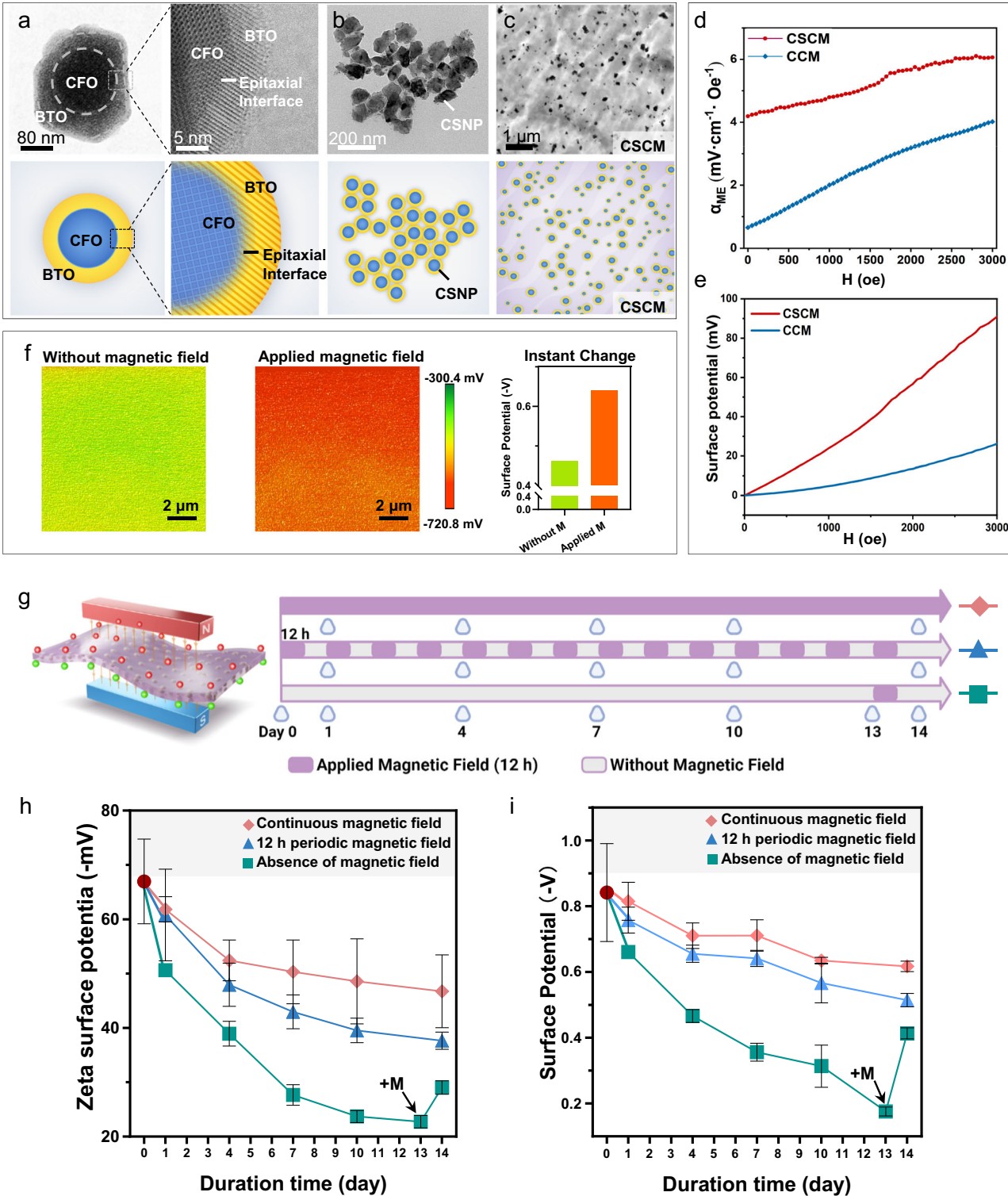

**Fig. 2 | Characterization of CFO@BTO core-shell particles and electrical properties of the CFO@BTO/P(VDF-TrFE) membranes. a** High-resolution transmission electron microscope (TEM) images and schematic illustration of the structure of CFO@BTO core-shell nanoparticles (CSNP). The enlarged view is intended to show the epitaxial relationship. **b** TEM image and schematic illustration of the CSNP. **c** Sectional TEM image and schematic illustration of the CFO@BTO core-shell composite membranes (CSCM). **d** Comparison of the magneto-electric coupling coefficient and the magnetic-field-induced surface potential **e** between the CSCM and CFO composite membranes (CCM). **f** Instantaneous change of surface potential before and after application of magnetic field (Scan size = 10 μm). **g** Schematic diagram of the time points for application of magnetic field and analysis of electrical properties. Created with BioRender.com. **h** Zeta potential ($n = 6$ independent membrane samples; mean ± SEM) and **i** surface potential measured by SKPM ($n = 5$ independent membrane samples; mean ± SEM) of CSCM under continuous exposure to magnetic field, 12 h periodic magnetic field and without magnetic field respectively. On the 13th day, a magnetic field was applied to the non-magnetic field exposed group, to observe its reactivation effect on the surface potential. Source data are provided as a Source Data file. At least three times of experiments were repeated independently.

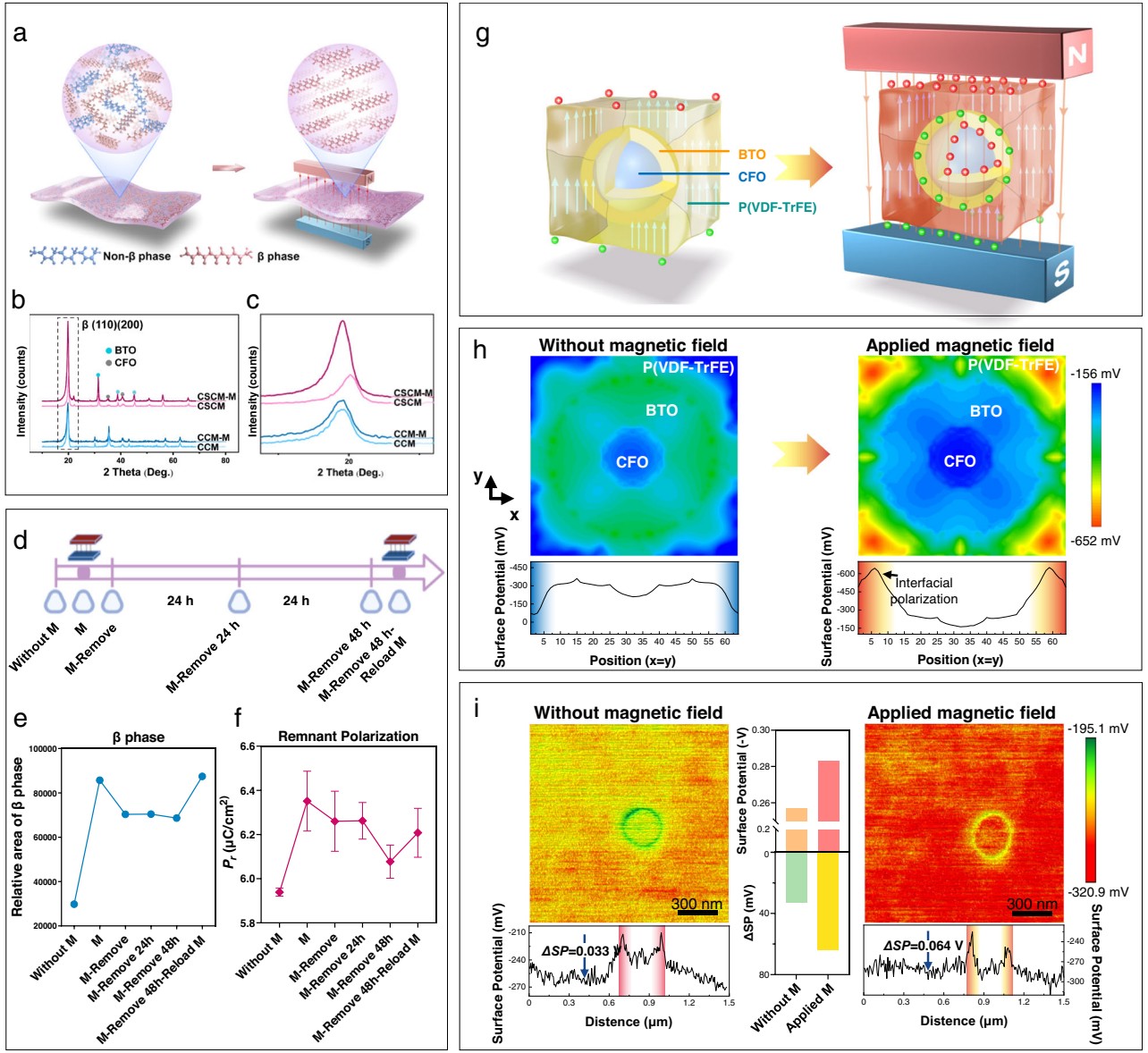

**Fig. 3 | Mechanisms of how magnetic field enhances polarization by phase transition and interfacial polarization. a** Schematic illustration of phase transition. **b** XRD results of CSCM and CCM, with or without exposure to the external magnetic field. **c** Enlarged image of β-phase change of XRD results. **d** Schematic diagram of the time points for detection of XRD and residual polarization intensity. Created with BioRender.com. **e** The semi-quantitative relative area of β-phase change, according to the loaded conditions of the external magnetic field. **f** The residual polarization intensity ($P_r$) of CSCM was changed according to the loaded conditions of the external magnetic field. ($n = 5$ independent membrane samples;

mean ± SEM) **g** Schematic diagram of interface polarization. **h** Phase-field simulation of surface potential change before and after magnetic field exposure. The surface potential values along the diagonal represent the interfacial polarization with the addition of a magnetic field. **i** Without (left) and with (right) magnetic field loading, the SKPM images of interface polarization and relative surface potential. Comparison of mean values of the surface potential within the detection range (scan size = 1.5 µm) and ΔSP, which represents the difference of surface potential between the interface and the matrix (ΔSP = $SP_{Interface}$ - $SP_{Matrix}$). Source data are provided as a Source Data file.

between CSNP and the P(VDF-TrFE) matrix[40]. To further verify the interfacial polarization, phase-field simulations of stress distribution, polarization level, and surface potential of CSCM were performed (Fig. 3h, Supplementary Fig. 7). The outcomes of phase-field simulation were consistent with the results of SKPM detection. When the magnetic field intensity increased, the stress distribution of CFO core, the polarization level of the BTO shell and the surface potential of P(VDF-TrFE) matrix increased significantly. This simulation reflected that the CSNPs received the magnetic force, and this force is converted to surface charges on the BTO shell. In the interfacial layer of the BTO shell and matrix P(VDF-TrFE), the increased charges on the BTO polarized the matrix P(VDF-TrFE), increased β-phase transition and surface potential of the membrane. The dual effect process refers to

force being transmitted from the magnetic field to the core-shell structure of the CSNP, which was then converted to interfacial polarization from the BTO shell to matrix P(VDF-TrFE).

To summarize, the dual effect can be defined by the following two relational expressions:

$$F \propto H, \lambda, E \tag{1}$$

$$P \propto F, d, 1/S \tag{2}$$

(The symbols within the formula denote as follows: $F$, stress; $H$, magnetic field intensity; $\lambda$, magnetostrictive coefficient; $E$, elastic

modulus; *P*, polarization intensity; *d*, piezoelectric coefficient; *S*, the unit contacting area)

## Evaluation of bone regeneration efficacy of CSCM in vitro and in vivo

To assess the biocompatibility of CSCM, the cell counting kit 8 (CCK8) assay, lactate dehydrogenase (LDH) assay, and live/dead cell staining (Supplementary Fig. 8a–c) were all performed. There were no significant effects of CSCM on BM-MSCs proliferation. Laser confocal microscopy and SEM imaging showed that BM-MSCs exhibited the largest spreading area on the CSCM surface, which indicated potent pro-osteogenic effects (Supplementary Fig. 9a, b)[41].

The expression of osteogenic-related marker genes *Runx2, BMP2, Sp7, Ocn, Opn, and Cola1* were further analyzed by real-time quantitative Polymerase Chain Reaction (RT-qPCR) at 3, 7, and 14 d[16]. All these osteogenic genes were significantly upregulated on CSCM-M (Fig. 4b). Western blot analysis of the osteogenic marker molecules, RUNX2, BMP2, and osteopontin (OPN), further confirmed the highest degree of osteogenic differentiation on CSCM-M (Fig. 4c). Immunofluorescence staining also displayed increased RUNX2 expression in BM-MSCs cultured on CSCM-M (Fig. 4a, Supplementary Fig. 10a)[42]. Alkaline phosphatase (ALP) staining (Supplementary Fig. 10c) and Alizarin red staining quantitative analysis (Supplementary Fig. 10d) revealed the highest osteogenic activity and most mineralized nodules on CSCM-M. These results thus demonstrated that CSCM-M possesses much osteoinductive capacity to enhance the osteogenic differentiation of BM-MSCs and promote bone defect repair.

To further assess the osteoinductive potential of CSCM with repeated magnetic field loading, we prepared cranial defect models in rats. Membranes were implanted covering the defects, and bone growth was evaluated after 4 and 8 weeks post-implantation (Fig. 4d). As evaluated by micro-computed tomography (Micro-CT), the CSCM-M group exhibited the best bone defect repair outcome (Fig. 4e, Supplementary Fig. 11a–d). Histological examination showed flat and contiguous new bone formation after implantation of CSCM with magnetic field loading (Fig. 4f, Supplementary Fig. 12a, b). Masson's Trichrome staining revealed the most mature osteoid tissue and abundant bone content within the defect area of the CSCM-M group (Supplementary Fig. 12c). These results thus confirmed that CSCM reactivated by periodic magnetic field loading can create a favorable electric microenvironment for bone regeneration in vivo.

## In situ repowering of CSCM enhances bone regeneration under co-morbidity conditions

To evaluate whether repowering of CSCM can reactivate bone regeneration under co-morbidity conditions, we constructed a cranial defect model in rats with bone osteogenesis repression being elicited by dexamethasone (Dex) or inflammation induced by lipopolysaccharide (LPS) respectively. For repression of osteogenesis with high-dose Dex injection[43], poorer periosteum formation and bone regeneration were confirmed at 7 d and 14 d respectively (Supplementary Fig. 13a, b). Then the osteogenesis repression rat model was used to evaluate the bone regenerative reactivation capacity of CSCM with periodic magnetic field loading (Fig. 5c). Membranes were implanted to cover the bone defect area followed by 3 days of Dex injection to induce repression of osteogenesis. From the 7th day onwards, the experimental group was treated with periodic magnetic field loading till the end of the 4th week. The micro-CT results (Fig. 5c, Supplementary Fig. 14a) showed the best osteogenesis in the CSCM-M group. The H&E staining and Masson's trichrome staining (Supplementary Fig. 14c, d) revealed the presence of newly formed dense bone, accompanied by bone trabeculae or vascular lumens. New bone formation was particularly conspicuous and prominent in the CSCM-M group. Bone volume /total volume (BV/TV) statistics (Supplementary Fig. 14b) confirmed the highest osteogenic efficacy in the CSCM-M group under osteogenic repression condition.

For the co-morbidity inflammation model, we implanted the composite membranes to cover the cranial bone defects and injected LPS for 3 days to induce systemic inflammation[44]. A significantly higher average white blood cell (WBC) count was determined in the LPS injected groups (Supplementary Fig. 15c). From the 7th day onwards, the experimental groups were treated with periodic magnetic field loading till the end of the 4th week. In Micro-CT, the CSCM-M group displayed the highest bone regeneration volume, which suggested that CSCM-M had the best pro-osteogenic effect under inflammatory conditions (Fig. 5d, Supplementary Fig. 15a, b). This effect was also confirmed through histological staining (Supplementary Fig. 15d, e). Contiguous and mature bone tissue filled the skull defects in the CSCM-M treated inflammatory models. Hence, these results show that the CSCM repowered by magnetic field loading can reactivate osteogenesis on demand for repairing bone defects. The bone regeneration of the CSCM-M group under co-morbidity conditions achieved the same repair effect as regular conditions. To further broaden the applications, we conducted experiments on rat mandibular defects under co-morbidity osteogenesis inhibition and systemic inflammation modeling. We compared the CSCM-M with commercial e-PTFE and collagen membranes. The experimental results demonstrate the efficacy of CSCM-M for mandibular defect repair compared with membranes used in clinic. (Supplementary Figs. 18–19).

Compared with previously developed stimuli-responsive biomaterials such as photothermal and sonodynamic materials[45], the CSCM-M displayed the highest in vivo osteogenic efficiency (Fig. 1a, Table S1, Table S2). Intramembranous bone formation starts as early as 3–7 days after injury in rats and lasts up to 28 days before bone remodeling[46–48]. In our study, at 4 weeks post-implantation, the osteogenesis efficiency of the CSCM-M groups reached 56% in the bone defect model and 52% in the bone defect combined with co-morbidity model. This meant that the volume of regenerated bone filled more than half of the defects in the first 4 weeks. The CSCM-M can thus restore normal osteogenesis within the bone defect with co-morbidity thereby mitigating the adverse healing conditions to be more similar to the regular bone defect healing process.

## Discussion

In this study, we fabricated a magnetic field-responsive core-shell structured CFO@BTO/P(VDF-TrFE) membrane. The membrane provides a remote stimuli-responsive electrical microenvironment that can reactivate bone regeneration even under inflammatory conditions or when osteogenesis is repressed.

Due to the epitaxial lattice structure and nano-sized confined space, most of the magnetic driving forces loaded on the core can be transferred to the shell layer[25,26]. The piezoelectric BTO shell can then convert the driving force to increased charge density on the shell surface. Because of the interfacial effect between the BTO shell and P(VDF-TrFE) matrix, the increased charge induces β-phase transition of the P(VDF-TrFE) matrix, resulting in increased surface potential of the CSCM. The dual effects of the increased interfacial polarization and enhanced β-phase transition would thus amplify the magneto-electric conversion efficiency. This effect accounts for the rechargeability of CSCM under magnetic field loading. The interfacial effect between the BTO nanoparticles and P(VDF-TrFE) matrix has thus been theoretically simulated[36–38]. To the best of our knowledge, this study provides straightforward experimental evidence for interfacial effect in magnetoelectric materials.

Magnetic field modulation is a safe and convenient method that can be easily applied with high tissue penetration depth in the clinic. Under inflammatory conditions or when osteogenesis is repressed, the regular clinical practice with conventional biomaterials is to

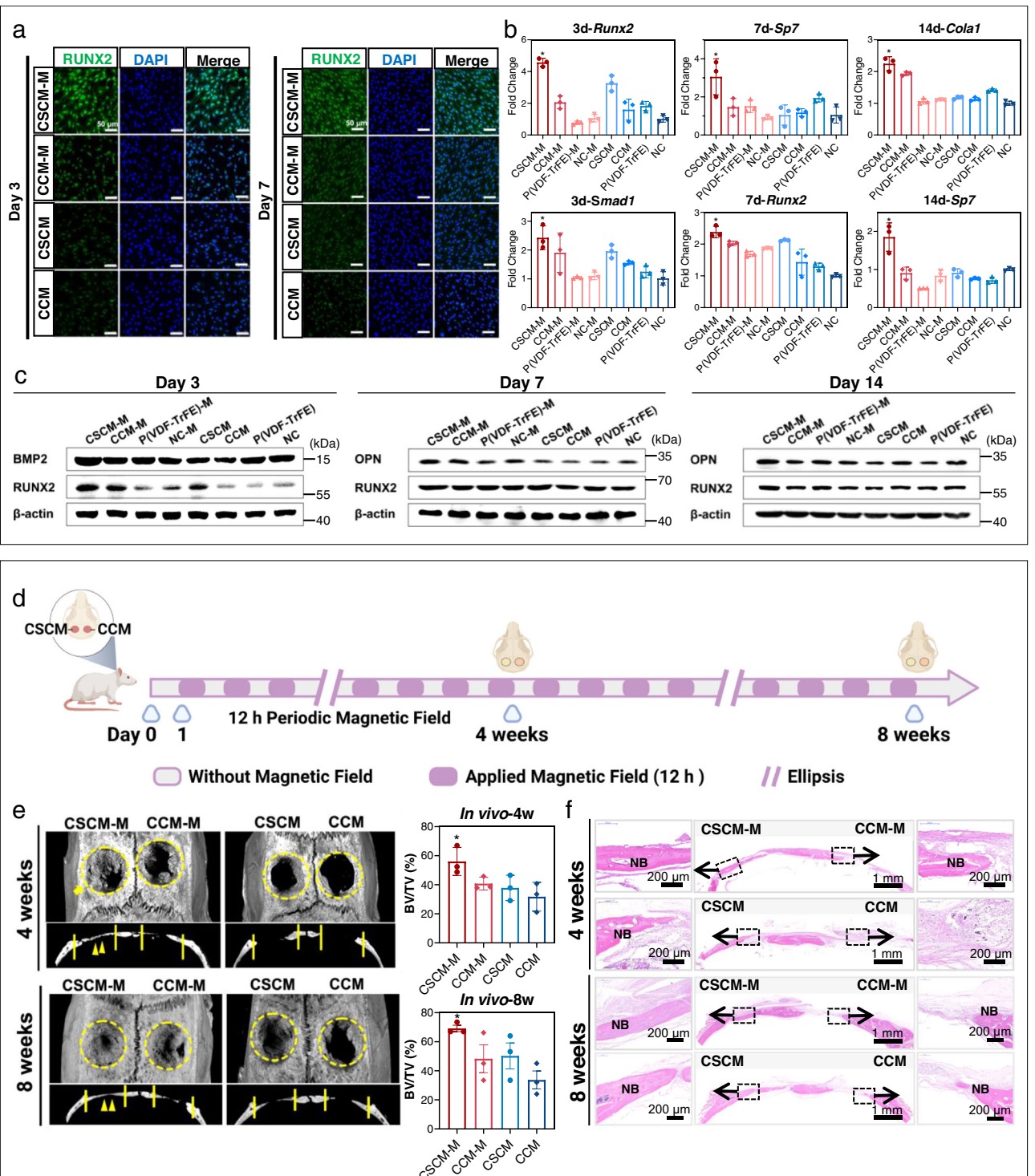

**Fig. 4 | Bone regeneration properties of CFO@BTO/P(VDF-TrFE) membranes in vitro and in vivo in a healthy state. a** Immunostaining images showed the upregulated expression of RUNX2 in the CSCM-M group after 3 and 7 days of cell culture (Scale bars: 50 μm). **b** Rt-qPCR analysis of upregulation of canonical osteogenic markers (*Runx2, Smad1*, Sp7, *Colα1*) in the CSCM-M group. *n* = 3 biologically independent samples; mean ± SEM.*P < 0.05, one-way ANOVA. **c** Western blot analysis revealed the upregulation of canonical osteogenic markers (BMP2,

RUNX2, OPN) in the CSCM-M group. **d** The timeline of magnetic field loading for the in vivo experiment. Created with BioRender.com. **e** Micro-CT images, bone volume statistics, and H&E staining of tissue sections **f** of bone defect repair after 4 and 8 weeks after CSCM and CCM implantation within skull defect in rats. NB new bone. *n* = 3 rats for per group and per time point; mean ± SEM. *P < 0.05, one-way ANOVA. Source data are provided as a Source Data file. At least three times of experiments were repeated independently.

immediately remove the implanted materials before bone healing completing, so as to ameliorate the co-morbidity conditions and avoid any further complications. With CSCM application, a smart in situ repowering strategy based on magnetic field loading can reactivate

bone regeneration under co-morbidity conditions. The CSCM can then be kept in the bone defect area until bone defect repair is completed. Hence, using stimuli-responsive magnetoelectric coupling membranes that can be repowered by an external magnetic field, opens up an

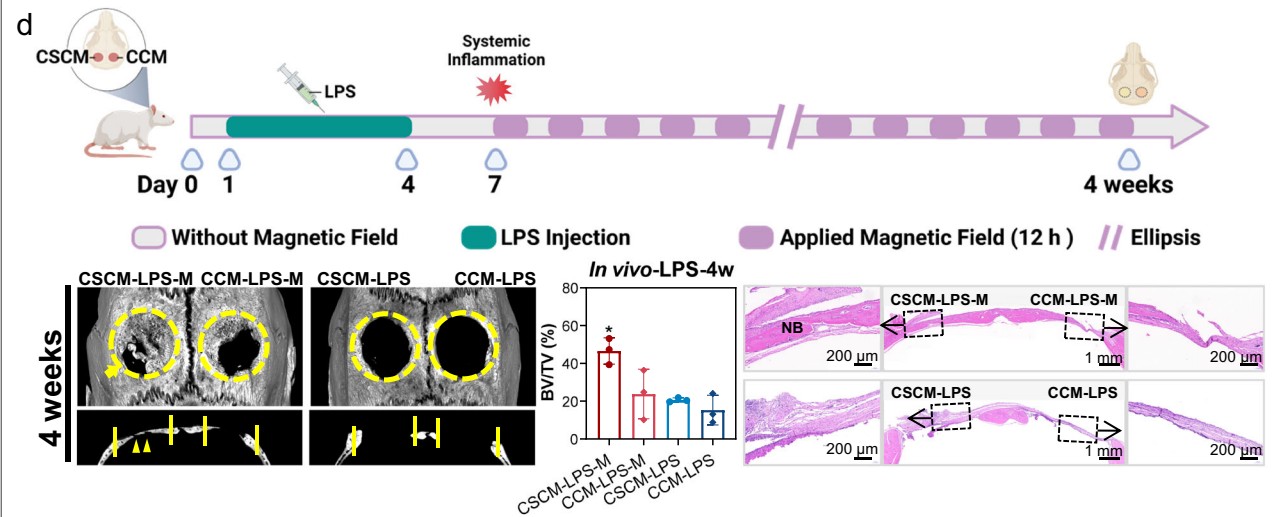

**Fig. 5 | Bone regeneration properties of CFO@BTO/P(VDF-TrFE) membranes in vivo under co-morbidity conditions. a**, **b** Schematic illustration of CSCM repairing skull defects under co-morbidity conditions, including the osteogenic inhibition model (**a**) and the systemic inflammatory model (**b**). **c** The process of Dexamethasone (Dex) injection for osteogenic suppression and application of magnetic field for CSCM reactivation. Micro-CT images and H&E staining of tissue sections after dexamethasone injection to inhibit osteogenesis, with and without magnetic treatment (4 weeks). NB new bone. $n = 3$ rats for per group; mean ± SEM. *P < 0.05, one-way ANOVA. **d** The schematic diagram of the systemic inflammation rat model and application of magnetic field for CSCM reactivation. Micro CT images and H&E staining of tissue sections after LPS injection for systemic inflammation, with and without magnetic treatment (4 weeks). $n = 3$ rats for per group; mean ± SEM. *P < 0.05, one-way ANOVA. Source data are provided as a Source Data file. The schematic diagrams were created with BioRender.com.

avenue for tissue regeneration and exhibits promising development potential for clinical applications.

## Methods

### Fabrication of the CFO@BTO core-shell nanoparticles

The modification of CFO nanoparticles with sodium oleate: 3 g CFO nanoparticles (Nanostructured & Amorphous Materials Inc) were dispersed in 40 ml sodium oleate (Aladdin) aqueous solution, followed by ultrasonic agitation for 1 h for dispersion. Then 1 mol/L hydrochloric acid was added slowly to adjust the pH to 5, stirring in the water bath at 60°C for 30 min. The suspension was then magnetically precipitated, rinsed with alcohol 3 times, and then dried at 60°C for 20 h to obtain CFO particles modified by sodium oleate.

Preparation of core-shell particles: Tetrabutyl titanate (Sigma-Aldrich) was dissolved in anhydrous ethanol (Aladdin) and stirred under ultrasonication for 30 min. The CFO nanoparticles modified by sodium oleate were dispersed into anhydrous ethanol. Then the two solutions were mixed and stirred for another 30 min. Barium acetate (Aladdin) was added during the ultrasonic stirring process, and acetic acid (Shanghai Macklin Biochemical Co.) was added slowly at a rate of 1 s per drop to adjust the pH value to 3–4, followed by the addition of distilled water at a rate of 3 s per drop. Then stirring was carried out within a 60°C water bath to form a sol, followed by stirring at 90°C to form gel. The gel was then transferred to petri dishes, dried in an oven at 120°C, and ground into powder after full drying. The samples were then placed into a muffle oven and heated to 500 °C for 6 h to remove the organic matter in the samples. After grinding again, baking was carried out for 2 h in a muffle oven at 800°C to optimize crystallization, and to obtain CFO@BTO core-shell nanoparticles.

### Fabrication of the CFO@BTO/P(VDF-TrFE) composite membranes

CFO@BTO core-shell nanoparticles were weighed according to the expected mass percentage (5%, 10%, 20%), which were then added into dimethylformamide (DMF, Sigma). The mixtures were placed in an ultrasonic device for 1 h to obtain an evenly dispersed suspension mixture. Then P(VDF-TrFE) powders ((70/30 mol % VDF/TrFE), Arkem, French) were added into the stirring mixture after the ultrasonic treatment. The suspension was dispersed by mechanical blending method and ultrasonic technique. After the blending process, the CFO@BTO/P(VDF-TrFE) suspension was spread on the glass substrate and dried at 55°C for 12 h. The membranes were then placed into an oven to anneal at 120°C for crystallization. For polarization, the membranes were treated with corona poling in a DC electric field of 15 kV for 60 min at room temperature. The method of fabricating 10% wt CFO/P(VDF-TrFE) membranes was described in the closely related study[23]. CFO nanoparticles were weighed and added into DMF. Then the mixtures were placed in an ultrasonic device for 1 h. After the ultrasonic treatment, P(VDF-TrFE) powders were added into the mixture. The solution was dispersed by the mechanical mixing method and ultrasonic technique. Then the follow-up processes and relevant parameters were exactly the same as the method of fabricating CSCM to ensure the consistency of sample synthesis.

### General characterization of nanoparticles and membranes

The morphology and structure of the membranes with different CFO@BTO core-shell nanoparticle weight content were observed under field emission-scanning electron microscopy (FE-SEM, S-4800, HITACHI, Japan). The morphology and lattice structure of core-shell nanoparticles were observed under high-resolution transmission electron microscopy (HRTEM, JEM-2100, JEOL, Japan). Membrane slicing was performed by ultra-thin slicing machine (Leica EM UC7, Germany) into slices with less than 200 nm thickness. Field emission transmission electron microscopy (FEI Tecnai G2 F30, USA) was used to observe the dispersion of core-shell particles within the membrane.

Chemical bonds within the membranes were analyzed by Fourier-transform infrared spectroscopy (FTIR, SP400, Norwalk, CT, USA). The phase structures were tested by X-ray diffraction spectroscopy (XRD, Rigaku D/max 2500VB2t/PC, Japan). During the detection process, a permanent magnet with strong magnetism was placed under the membranes to construct a magnetic field environment. The surface roughness of the membranes was analyzed by atomic force microscopy (AFM) (Bruker, Santa Barbara, CA, USA) in the contact mode with a scanning size of 500 nm square. The water contact angle was measured by a video contact angle instrument (KRÜSS, Germany). The mechanical properties were examined using a universal mechanical machine (INSTRON-1121, USA) at a strain rate of 30 mm min$^{-1}$. The samples ($n = 5$) were cut into 70 mm long, 3.5 mm wide, and 0.04 mm high pieces. All testing was carried out at room temperature and under dry conditions.

### Electrical characterization of membranes

Zeta surface potential of membranes in every group were measured using a Zeta Sizer Nano-ZS Instrument (Malvern Instruments, Worcestershire WR, UK) at room temperature. The surface potential of the membranes was examined by Kelvin probe force microscopy (KPFM, Bruker) in the Electrical & Magnetic lift mode. A DC magnetic field within a range of about 0 to 4000 oe was generated through a power source that can generate DC currents in the range of 0 to 5 A. The scanning data were collected by the software NanoScope 9.1, and the results were analyzed using a software NanoScope Analysis 1.5. There were 6 groups in the experiments for testing zeta potential and the surface potential. The membranes in all groups were always soaked within a petri dish with PBS. Group 1 - CSCM were kept in the magnetic field. Group 2 - the CSCM were not in the magnetic field all the time, and the magnetic field was added for 12 h on day 13. Group 3 - the CSCM were not in the magnetic field for 12 h, and the magnetic field was loaded for 12 h, and then the cycle was repeated. Group 4 - the CSCM were not in the magnetic field for 24 h, and the magnetic field was loaded for 12 h, and then the cycle was repeated. Group 5 - the CSCM were not in the magnetic field for 48 h, and the magnetic field was loaded for 12 h, and then the cycle was repeated. Group 6 - the CSCM were not in the magnetic field for 72 h, and the magnetic field was loaded for 12 h, and then the cycle was repeated.

Copper electrodes with diameter of 3 mm were sputtered on both sides of the prepared nanocomposites membranes for P-E loop measurement. Bipolar polarization–electric feld (P–E) hysteresis loops measurements were performed on the polarization loop & dielectric breakdown test system (PolyK Technologies, LLC) at a frequency of 10 Hz. During the P-E loop measurements, a permanent magnet with strong magnetism was placed under the glass container containing silicone oil to create a magnetic field environment. Time points of detection: without magnetic field, applied magnetic field, the magnetic field removed on the spot, 24 h after the magnetic field was removed, 48 h after the magnetic field was removed, and reloading of magnetic field.

The magnetoelectric coupling coefficient α was determined using a multiferroic material magnetoelectric measuring system (Super ME-II, Quantum Design, USA). The test and calculation were performed according to published studies[22,24]. The membrane surfaces were coated with silver paste (Ted Pella, USA) and then connected to the magnetoelectric measuring system. Then, a direct current (dc) magnetic field (0-3000 Oe) and an alternating current (ac) magnetic field (0.82 Oe, 5 kHz) were applied. The magnetic-field-induced surface potential V was calculated by Eq. (3) and Eq. (4):

$$\alpha = \Delta X / (H_{ac} \times d) \tag{3}$$

$$V = \alpha \times d \times H_{dc} \tag{4}$$

The magnetoelectric signal voltage was represented by $\Delta X$, with the ac and dc magnetic field intensity being represented by $H_{ac}$, $H_{dc}$, and the thickness of the CFO/P(VDF-TrFE) membranes being represented by d.

## Phase-field simulations

In the phase-field simulation, the polarization $P_i(r,t)$ is the order parameter. The spatial-temporal evolution of the polarization can be described by the time-dependent Ginzburg-Landau equation (TDGL) as follows:

$$\frac{\partial P_i(\boldsymbol{r},t)}{\partial t} = -L\frac{\delta F}{\delta P_i(\boldsymbol{r},t)}, (i=1,2,3) \tag{S1}$$

where $L$ is the kinetic coefficient, and $F$ is the total free energy of the system, which is expressed as,

$$F = \iiint (f_{bulk} + f_{elastic} + f_{electric} + f_{grad})dV \tag{S2}$$

where $V$ is the system volume. The bulk energy density $f_{bulk}$ can be calculated by:

$$
\begin{aligned}
f_{bulk} = {} & \alpha_1(P_1^2 + P_2^2 + P_3^2) + \alpha_{11}(P_1^4 + P_2^4 + P_3^4) \\
& + \alpha_{12}(P_1^2 P_2^2 + P_1^2 P_3^2 + P_2^2 P_3^2) + \alpha_{112}[P_1^4(P_2^2 + P_3^2) \\
& + P_2^4(P_1^2 + P_3^2) + P_3^4(P_1^2 + P_2^2)] + \alpha_{111}(P_1^6 + P_2^6 + P_3^6) \\
& + \alpha_{123} P_1^2 P_2^2 P_3^2
\end{aligned}
\tag{S3}
$$

where $P_1$, $P_2$, $P_3$ are the polarization components. $\alpha_1$, $\alpha_{11}$, $\alpha_{12}$, $\alpha_{111}$, $\alpha_{112}$ and $\alpha_{123}$ are Landau coefficients. The elastic energy density can be expressed as follows:

$$f_{elastic} = \frac{1}{2} c_{ijkl} e_{ij} e_{kl} = \frac{1}{2} c_{ijkl}(\varepsilon_{ij} - \varepsilon_{ij}^0)(\varepsilon_{kl} - \varepsilon_{kl}^0) \tag{S4}$$

where $c_{ijkl}$ is the elastic stiffness constant, $e_{ij}$ is the elastic strain, $\varepsilon_{ij}$ is the total strain, and $\varepsilon_{ij}^0$ is the eigenstrain (electrostrictive stress-free strain). Using the cubic phase as the reference, $\varepsilon_{ij}^0$ can be calculated by $\varepsilon_{ij}^0 = Q_{ijkl} P_k P_l$, where $Q_{ijkl}$ is the electrostrictive coefficient. The gradient energy density can be expressed as follows:

$$f_{grad} = \frac{1}{2} G_{ijkl} \frac{\partial P_i}{\partial r_j} \frac{\partial P_k}{\partial r_l} \tag{S5}$$

where $G_{ijkl}$ is the gradient energy coefficient. The electrostatic energy density is expressed as:

$$f_{electric} = -\frac{1}{2} \varepsilon_0 K_{ij}^b E_i E_j - E_i P_i \tag{S6}$$

where $K_{ij}^b$ is the background relative permittivity and $E_i$ is the electric field, which can be calculated from the following equation:

$$E_i = -\frac{\partial \varphi}{\partial r_i} \tag{S7}$$

The electric potential $\varphi$ can be obtained from the electrostatic equilibrium equation:

$$\varepsilon_0 K_{ij}^b \frac{\partial^2 \varphi}{\partial r_i \partial r_j} = -\frac{\partial P_i}{\partial r_i} \tag{S8}$$

Equations are numerically computed by a semi-implicit Fourier-spectral method[49]. The effect of an external applied magnetic field $H_{ex}$

on the system energy can be taken into account through the interaction between magnetization and the external field[50].

$$E_{external} = -\mu_0 M_s \int H_{ex} m dV \tag{S9}$$

For a cubic magnetostrictive material, the deformation associated with the local magnetization is described by the stress-free strain:

$$\varepsilon_{i,j}^0 = \begin{cases} \frac{3}{2}\lambda_{100}(m_i^2 - \frac{1}{3}) & (i=j) \\ \frac{3}{2}\lambda_{111} m_i m_j & (i \neq j) \end{cases} \tag{S10}$$

where $\lambda_{100}$ and $\lambda_{111}$ are the magnetostrictive constants of a cubic crystal. For $CoFe_2O_4$, $\lambda_{100} = -5.9 \times 10^{-4}$, $\lambda_{111} = 1.2 \times 10^{-4}$.

The Landau coefficients for the calculation of $BaTiO_3$ and PVDF-TrFE are shown in Table 1 and Table 2. The simulation size is $64\,\Delta x \times 64\Delta x \times 64\Delta x$, the particle diameter is $50\Delta x$, and the grid space in real space is $\Delta x = 1.0$ nm.

## Cell culture

Rat BM-MSCs (Cyagen Bioscience Inc., China) were cultured in α-MEM (HyClone, USA, SH30265.01B) supplemented with 10% (v/v) fetal bovine serum (FBS) (Gibco 10099141) and 1% (v/v) penicillin-streptomycin solution (Gibco 15140122) within a cell culture incubator under standard conditions (37°C, 5% $CO_2$). The culture medium was refreshed every 2 days.

## Proliferation and toxicity of BM-MSCs

Rat BM-MSCs ($1 \times 10^5$ cells $mL^{-1}$) were seeded onto the CFO@BTO/P(VDF-TrFE) membranes with different proportions of CFO@BTO nanoparticles in 6-well plates. To analyze cell proliferation, the medium of the BM-MSCs seeded in 6-well plates was replaced with culture medium containing 10% CCK8 kit (Dojindo, Shanghai China) solution after seeding for 24, 48, and 72 h, followed by incubation at 37 °C for an additional 2 h. The supernatant was then placed into a 96-well plate, and the absorbance was then measured using a microplate reader at 450 nm, with 3 replicates per group. Cell toxicity was assayed using a LDH kit (Beyotime Biotechnology Inc., Shanghai China) at 24, 48, and 72 h of culture, with absorbance measurements being read at a wavelength of 490 nm, using an enzyme-linked immunosorbent assay reader (Bio-Rad, Hercules, CA, USA). To analyze cellular viability, a Live/Dead assay was performed with calcein AM and ethidium homodimer (Invitrogen).

## Spreading of BM-MSCs

BM-MSCs were cultured on the membranes for 24 h, and then the cells were rinsed 3 times with PBS precooled at 4°C and fixed with 2.5% (w/v) glutaraldehyde at 4°C for 2 h. After abandoning the fixed solution, the membranes and cells were soaked in 0.18 mol/L sucrose at 4°C for 2 h. Dehydration with ethanol was carried out in a gradient of 30%, 50%, 70%, 80%, and 90%, at 4°C for 10 min each. Dehydration with anhydrous ethanol was carried out 3 times, 10 min each. After $CO_2$ critical point drying, a gold spraying operation was carried out. The images were acquired using FE-SEM (S-3000N, Japan)

## ALP activity assay

For ALP staining, BM-MSC ($1 \times 10^5$ cells $mL^{-1}$) were cultured on the membranes. After 3 and 7 days of culture, the cells were washed in phosphate-buffered saline (PBS) three times and then lysed by cell lysis buffer without inhibitors (P0013J, Beyotime, China), for Western blot and IP assays. The BM-MSC lysates were collected and tested by an Alkaline Phosphatase Assay Kit (P0321, Beyotime), with ALP activity measurements being carried out at 405 nm. The relative normalized alkaline phosphatase activity was expressed as per optical density (OD) value of total protein for each sample.

**Table 1 | The Landau coefficients for the calculation of BaTiO₃[51]**

| $\alpha_1$ ($C^{-2}$ $m^2$ N) | $4.124 \times (T-115) \times 10^5$ |
|---|---|
| $\alpha_{11}$ ($C^{-4}$ $m^6$ N) | $-2.097 \times 10^8$ |
| $\alpha_{12}$ ($C^{-4}$ $m^6$ N) | $7.974 \times 10^8$ |
| $\alpha_{111}$ ($C^{-6}$ $m^{10}$ N) | $1.294 \times 10^9$ |
| $\alpha_{112}$ ($C^{-6}$ $m^{10}$ N) | $-1.950 \times 10^9$ |
| $\alpha_{123}$ ($C^{-6}$ $m^{10}$ N) | $-2.500 \times 10^9$ |
| $\alpha_{111}$ ($C^{-8}$ $m^{14}$ N) | $3.963 \times 10^{10}$ |
| $\alpha_{1112}$ ($C^{-8}$ $m^{14}$ N) | $2.529 \times 10^{10}$ |
| $\alpha_{1122}$ ($C^{-8}$ $m^{14}$ N) | $1.637 \times 10^{10}$ |
| $\alpha_{1123}$ ($C^{-8}$ $m^{14}$ N) | $1.367 \times 10^{10}$ |

**Table 2 | The Landau coefficients for the calculation of PVDF-TrFE[52]**

| $\alpha_1$ ($C^{-2}$ $m^2$ N) | $1.412 \times (T-42) \times 10^7$ |
|---|---|
| $\alpha_{11}$ ($C^{-4}$ $m^6$ N) | $-1.842 \times 10^{11}$ |
| $\alpha_{111}$ ($C^{-6}$ $m^{10}$ N) | $2.585 \times 10^{13}$ |

## Alizarin red S staining

Assessment of mineralization level was determined by alizarin red staining. BM-MSC ($1 \times 10^5$ cells $mL^{-1}$) were cultured on the membranes. After 14 and 21 days of culture, cells were fixed in 4% (w/v) paraformaldehyde for 20 min. Staining with 1% (w/v) Alizarin Red S pH 4.2 (Sigma- Aldrich) was used to detect calcium deposition. After incubation for 30 min at room temperature, the Alizarin Red S stain within the samples were dissolved in 10% cetylpyridinum chloride (Sigma) for quantitative analysis. Three replicate absorbance readings for each group was measured at 562 nm.

## Immunofluorescence analysis

Cells cultured on membranes were fixed in 4% (w/v) paraformaldehyde for 15 min. After rinsing three times with PBS, 0.1% (w/v) Triton X-100 (diluted with PBS) was used to permeabilize the samples for 10 min. Then, the cells were blocked with 3% (w/v) bovine serum albumin (BSA; diluted with PBS) for 1 h at room temperature. The samples were then incubated with the primary antibody - polyclonal rabbit anti-RUNX2 (1:100, diluted with 3% (w/v) BSA solution; Abcam, ab114133) overnight at 4°C. Then the primary antibody was removed, followed by washing three times with PBS. The goat anti-rabbit IgG H&L Alexa Fluor 488 (1:500, diluted with 1 wt% BSA solution; Abcam) was utilized as a secondary antibody, and incubated with the sample for 1 h in darkness. After that, 4′,6-Diamidino-2-phenylin- dole (1:1000, diluted in PBS solution, DAPI, Sigma) was used to stain the cell nuclei. Phalloidin (Solarbio) was used for cytoskeletal staining. Images were captured under laser-scanning confocal microscopy (Lecia) and analyzed with LAS X Software (Media Cybernetics).

## Western blot analysis

The total protein contents of cultured cells were extracted in RIPA lysis buffer (Beyotime, Shanghai, China) with a protease inhibitor cocktail (Thermo Fisher Scientific, Wilmington, DE, USA) on ice. The protein concentration of each sample was quantified using a BCA protein assay kit (Beyotime). Six times SDS Sample Loading Buffer (P0015F; Beyotime) was added into the protein extract before heating at 100 °C for 5 min. The total protein extract (40 μg) was then separated by 10% (w/v) sodium dodecylsulfate polyacrylamide gel electrophoresis. Then the separated proteins were transferred to a PVDF membrane. The membranes were blocked with 5% (w/v) skimmed milk and incubated with the primary antibody at 4 °C overnight. Then the membranes were rinsed three times with Tris-buffered saline/Tween 20 (TBST), followed by incubation with a secondary antibody conjugated with horseradish peroxidase (HRP) for 1 h at room temperature. The immunoreactive protein bands were visualized with an eECL Western Blot Kit (CoWin Bio., Jiangsu, China) on a film exposure machine. The primary antibodies: anti-RUNX2 (ab76956, 1:1000, diluted with 5% w/v skimmed milk), anti-BMP2 (ab214821, 1:1000, diluted with 5% w/v skimmed milk) and anti-Osteopontin (ab63856, 1:1000, diluted with 5% w/v skimmed milk) were purchased from Abcam. The primary antibody anti-β-actin (AF0003, 1:1000, diluted with 5% w/v skimmed milk) and secondary antibody HRP-labeled IgG (A0208, A0216, 1:1000, diluted with TBST) were purchased from Beyotime, China. β-Actin was utilized as the internal control. Protein levels were analyzed by using an Image J analysis software.

## Real-time quantitative PCR analysis

Total RNA was extracted using TRIzol Reagent (15596026, Invitrogen, USA) according to the manufacturer's instructions. The quality and quantity of the RNA samples obtained were subjected to spectrophotometric analysis using a bio-photometer (Thermo Scientific™ NanoDrop8000). The RNA was then reverse transcribed into complementary DNA (cDNA) using a Reverse Transcription kit (RR037A, Takara Bio Inc., Japan). Quantitative real-time polymerase chain reaction (qPCR) was performed using a FastStart Universal SYBR Green Master Mix (Rox) system with QuantStudio Design & Analysis Desktop Software (Thermo Fisher Scientific). The primer sequences are shown in Table 3. Glyceraldehyde-3-phosphate dehydrogenase (Gapdh) was utilized as the internal control.

## Rat calvarial defect repair in vivo

All animal experiments were approved by the Animal Care and Use Committee of Peking University (IACUC number: LA2021230).

7-week-old male Sprague-Dawley (SD) rats were used in this study. The experimental protocol was approved by the Animal Care and Use Committee of Peking University (IACUC number: LA2021230). Before the dorsal cranium was surgically exposed, the rats were anesthetized with phenobarbital sodium (100 mg/kg) via intraperitoneal injections. Then two full-thickness bone defects (5 mm diameter) were prepared on each side of the parietal bone to establish the cranial defect model.

There were 6 groups: Group 1, the left defect was covered with CSCM and the right defect was covered with CCM. Group 2, the left defect was covered with CSCM and the right defect was covered with polarized P(VDF-TrFE) membranes. Group 3, the left defect was covered with CCM and the right defect was not covered as a negative control. Group 4–6 were the same as Group 1–3. Group 1-3 were kept in custom-made cages with magnets on a 12-hour daily cycle. Group 4–6 were raised in conventional cages.

At 4 and 8 weeks after membrane implantation, SD rats were euthanized by asphyxiation with $CO_2$, followed by skull explantation. Samples were fixed in 4% (w/v) paraformaldehyde for 24 h at room temperature. The specimens were examined using a micro-CT scanner (Inveon Multi Modality Gantry-STD, Siemens, USA), using the reconstruction software Cobra, and analyzed with 3D post-processing workstation Inveon Research Workplace V 2.2.0.

After the micro-CT scanning, the samples were decalcified with EDTA decalcification solution and embedded in paraffin. Histomorphology staining and analysis were performed on 5-μm-thick histology sections of the central portion of the skull defect. The sections were subjected to hematoxylin and eosin (H&E) and Masson's trichrome staining, according to the manufacturer's protocols. Images were captured using an Olympus D70 camera mounted on a Nikon Eclipse E800 microscope.

**Table 3 | Primer sequences used for quantitative real-time PCR analysis**

| Target gene | Forward sequence (5'–3') | Reward sequence (5'–3') |
|---|---|---|
| Runx2 | CAGTATGAGAGTAGGTGTCCCGC | AAGAGGGGTAAGACTGGTCATAGG |
| Smad1 | CCCCAACAGCAGCTACCCCAACTC | TGGGCCATGGGGTCTTCAGGAG |
| Sp7 | CTGGGAAAAGGAGGCACAAAGA | GGGGAAAGGGTGGGTAGTCATT |
| Cola1 | AGAGGCATAAAGGGTCATCGTG | AGACCGTTGAGTCCATCTTTGC |
| Gapdh | CTGGAGAAACCTGCCAAGTATG | GGTGGAAGAATGGGAGTTGCT |

**Osteogenic repression by high dose of dexamethasone in vivo**

After anesthesia and surgical membrane implantation, the rats were injected intramuscularly with dexamethasone at a dose of 1 mg/kg for 3 consecutive days from the first day after surgery. Samples collected at 1 week were treated with H&E staining and Masson's trichrome staining to detect osteogenic inhibition. Similarly, micro-CT and histological staining at 2 weeks were used to confirm the osteogenic inhibition model.

The rats were raised in custom-made cages with magnets from 7 days after operation and were subjected to a 12-hour daily cycle for 4 weeks. Other steps and groups were the same as the above animal experiments.

**Systemic inflammatory animal models used for osteogenesis experiments in vivo**

After anesthesia and surgical membrane implantation, the rats were treated with LPS (Solarbio, L8880) at a dose of 1 mg/kg from 1 to 4 days by intraperitoneal injection. Tail vein blood sampling was performed at 1 week for routine blood testing and white blood cell count. Then the rats were housed in custom-made cages with magnets and subjected to a 12-hour daily cycle for 4 weeks. Other steps and groups were the same as the above animal experiments.

**Rat mandibular defects repair in vivo**

The rats were anesthetized with phenobarbital sodium (100 mg/kg) via intraperitoneal injections. The rat cheek skin was cut horizontally. Blunt separation of mandibular muscles to expose the mandibular angle. At the junction of the mandibular body and the mandibular ramus, the penetrating bone defects with a diameter of 3 mm were made at two side respectively. The membranes of each group were cut into suitable size and implanted to cover the defect area. After sterile saline irrigation, 5–0 absorbable sutures were used to reset the muscle and 4-0 sutures were used to close the skin. The co-morbidity model of osteogenesis inhibition and systemic inflammation were prepared in the same process as previously mentioned.

There were 6 groups: Group 1, the right defect was covered with CSCM and the left defect was covered with collagen membrane (Dentium, Korea). Group 2, the right defect was covered with CSCM and the left defect was covered with e-PTFE membranes (Imedcare, China). Group 3, the right defect was covered with CSCM and the left defect was not covered as a negative control. Group 4-6 were the same as Group 1-3. Group 1-3 were kept in custom-made cages with magnets on a 12-hour daily cycle. Group 4−6 were raised in conventional cages.

The rats were raised in custom-made cages with magnets for 4 weeks. Other steps were the same as the above animal experiments.

**Statistical analysis and schematic diagram**

The data were presented as the mean ± standard deviation (SD). Each experiment was repeated at least three times. Statistically significant differences ($P < 0.05$) were measured using one-way analysis of variance (ANOVA) combined with the Student-Newman-Keuls (SNK) multiple comparison post hoc test. Statistical analysis was performed using SPSS-21.0 (International Business Machines Corporation (IBM), USA) software. GraphPad Prism Version 8.2 was used to realize data visualization. Part of the schematic icons were created with BioRender.com with a confirmation of publication and licensing rights.

**Reporting summary**

Further information on research design is available in the Nature Portfolio Reporting Summary linked to this article.

## Data availability

All relevant data supporting the key findings of this study are available within the article and its Supplementary Information files. Source data are provided with this paper.

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

## Acknowledgements

We acknowledge support from the National Key R&D Program of China (2018YFE0194400, W.L.; 2021YFC2400403, X.D.; 2021YFC2400404, L.C.; 2021YFB3800800, X.Z.), National Natural Science Foundation of China (No. 81600905, W.L.; No. 81991505, X.D.; No. 82022016, X.Z.; No. 82101067, C.Z.; No. U22A20314, W.L.; No. U22A20160, X.D.), National Science and Technology Basic Resources Project (2018FY101004, M.X.).

## Author contributions

W.L., H.Z., C.Z., L.C., and X.D. conceived the experiment. W.L., H.Z., S.X., and H.H. performed and analyzed the Phase-field simulation. W.L., H.Z., C.Z., Fe.Z., L.W., Fa.Z., Yi.C., and Yu.C. synthesized and characterized the materials and performed the in vitro and in vivo experiments. Y.Hu., M.X., and Y.He commented on the devices used. J.Z., H.H., and Y.S. commented on materials and experiment designs. W.L., H.Z., X.D. wrote the manuscript. H.Z. prepared figures. W.L., J.Z., H.H., Y.S., B.C., H.Z., and X.D. revised the manuscript.

## Competing interests

The authors declare no competing interests.

## Additional information

[1]Department of Geriatric Dentistry, Peking University School and Hospital of Stomatology & National Center for Stomatology & National Clinical Research Center for Oral Diseases & National Engineering Research Center of Oral Biomaterials and Digital Medical Devices, No.22, Zhongguancun South Avenue, Haidian District, Beijing 100081, P. R. China. [2]Hospital of Stomatology, Guanghua School of Stomatology, Sun Yat-sen University, Guangzhou, P. R. China. [3]School of Materials Science and Engineering & Advanced Research, Institute of Multidisciplinary Science, Beijing Institute of Technology, Beijing, P. R. China. [4]Department of Orthopedics, The Second Xiangya Hospital, Central South University, Changsha, P. R. China. [5]Third Clinical Division, Peking University School and Hospital of Stomatology & National Center for Stomatology & National Clinical Research Center for Oral Diseases & National Engineering Research Center of Oral Biomaterials and Digital Medical Devices, Beijing, P. R. China. [6]First Clinical Division, Peking University School and Hospital of Stomatology & National Center for Stomatology & National Clinical Research Center for Oral Diseases & National Engineering Research Center of Oral Biomaterials and Digital Medical Devices, Beijing, P. R. China. [7]Central Laboratory, Peking University School and Hospital of Stomatology & National Center for Stomatology & National Clinical Research Center for Oral Diseases & National Engineering Research Center of Oral Biomaterials and Digital Medical Devices, Beijing, P. R. China. [8]Department of Physics, Beijing Normal University, Beijing, P. R. China. [9]State Key Laboratory of New Ceramics and Fine Processing Department of Materials Science and Engineering Tsinghua University, Beijing, P. R. China. [10]Department of Dental Materials & Dental Medical Devices Testing Center, Peking University School and Hospital of Stomatology & National Center for Stomatology & National Clinical Research Center for Oral Diseases & National Engineering Research Center of Oral Biomaterials and Digital Medical Devices, Beijing, P. R. China. [11]Department of Stomatology, Union Hospital, Tongji Medical College, Huazhong University of Science and Technology, Wuhan, P. R. China. [12]These authors contributed equally: Wenwen Liu, Han Zhao, Chenguang Zhang. ✉e-mail: zhangxuehui@bjmu.edu.cn; hbhuang@bit.edu.cn; chenlili1030@hust.edu.cn; kqdengxuliang@bjmu.edu.cn

