## [Peer Review File · Nature Communications]

Editorial Note: This manuscript has been previously reviewed at another journal that is not operating a transparent peer review scheme. This document only contains reviewer comments and rebuttal letters for versions considered at *Nature Communications* .

REVIEWER COMMENTS

Reviewer #1 (Remarks to the Author):

The work by Deng et al. presents a novel strategy to induce bone regeneration in full sized skull defects by applying a flexible magnetoelectric membrane on top of the defect and stimulating with a magnetic field. The work analyses in a rat model the effect of a pre-charged membrane alone and an even more pronounced magnetoelectric effect within a magnetic field. To verify a clinical relevance, different challenged pre-clinical model systems were compared against a “WT” setting using dexamethasone or a general inflamed setting.

The manuscript is largely well written and describes in my eyes a novel strategy and concept in sufficient detail. I have, however, some major concerns that need to be addressed prior to acceptance:

- the authors target an osteogenic efficiency (whatever this is) and give concrete numbers in Fig. 1 in a scale. I would suggest removing these numbers and clarifying in the Figure caption what you define as “osteogenic efficiency”.
- can you explain why the osteogenic stimulation in the “WT” is comparable to the one in the “impaired” dexa group?
- Why is in Figure 2 the curve of the simple membrane increasing at day 14 after I continuous decline until then?
- In Figure 4 I would suggest for BV/TV similar scales to ease comparison – same applies for the other graphs in Figure 5.
- all groups are compared against the “WT” setting of such defects but the endogenous potential of healing in the non-treated is quite well. To allow for an estimation of the clinical potential, a control group that employs a clinical used therapy would be helpful.

Reviewer #2 (Remarks to the Author):

In this manuscript, the authors presented a strategy for in situ on-demand activation of flexible magnetoelectric membrane that was shown to enhance bone defect repair. For bone defect repair

under co-morbidity conditions, the use of biomaterials that can be non-invasively regulated is highly desirable to avoid further complications and to promote osteogenesis. However, there are some challenges in clinical applications to achieve efficient osteogenesis with stimuli-responsive materials. The authors claimed to have developed polarized CoFe₂O₄@BaTiO₃/P(VDF-TrFE) core-shell composite membranes (CSCM) with high magnetoelectric conversion efficiency for activating bone regeneration on demand. An external magnetic field force conducted on the CoFe₂O₄ core could increase charge density on the BaTiO₃ shell and strengthens the β -phase transition in the P(VDF-TrFE) matrix. This energy conversion increased the membrane surface potential, which hence activated osteogenesis. In vivo experiments confirmed that repeated applications of external magnetic field on CSCM can enhance bone defect repair, even when osteogenesis repression is elicited by dexamethasone or lipopolysaccharide-induced inflammation.

Overall, this study provides some insights into smart biomaterials for activating osteogenesis in vivo, which is generally interesting to the field although there exist significant issues that shall be taken care of by the authors. At this stage, the suitability of the manuscript for the target journal is questionable. Detailed comments follow below:

- The phrasing is inaccurate in many places. For example, they termed the membranes “core-shell composite membranes (CSCM)”, which is not correct. The particles were core-shell not the membranes. The membranes were only encapsulating these core-shell particles yet the membranes themselves were not core-shell. The language is highly misleading. In fact, how the particle parameters and membrane parameters were optimized remains elusive. How do they determine the amount of particles within the membranes? Why were the specific sizes of the core/shell chosen? How was the membrane thickness selected? None of these parameters were carefully examined by the authors leaving the readership only in perplex without understanding the rationale behind the fundamentals of the study.
- Similarly, they said TEM shows the cross-sectional views of the particles, which is incorrect. TEM shows projection views not necessarily a layer or surface of the cross-section.
- The wording ‘on-demand’ is also very problematic. From the study designs, it was not used ‘on-demand’ but rather programmed. It seems that magnetic field was applied for 12 h then off for 12 h and cycled for 14 days or 28 days. There was nothing on-demand. On-demand means that they could have the membrane implanted, and only activates it at a certain time point oftentimes not from the beginning of implantation. For example, can they keep the membranes there for some days without activation and then activate to induce bone regeneration, and compare with the group without any activation to see how the delayed activation may ‘on-demand’ improve osteogenesis? Or can the membrane be pre-implanted, followed by injury, then activation of the membrane to induce osteogenesis ‘on-demand’?
- Related to the last concern, the degradation or stability profiles of these membranes remain elusive. Likely the membranes would not quite degrade in vivo. Yet how is their stability in terms of water uptake, or more importantly, surface adsorption of proteins/other molecules that may significantly change their charging profiles? The authors should carefully study the long-term membrane properties

including activation abilities in physiological media that are rich in body fluid molecules to explain these questions and benefit the readership.

- Are the membranes truly rechargeable? For actual experiments the authors used 12/12 h magnetic field for up to 4 weeks, but in Figure 3 they only showed a couple polarization cycles. Upon M-removal the peak was only dropping slightly still maintaining a high level and then upon M-reload it went up to the peak. How about more cycles? How many cycles can they do this? If they used 12/12 h cycles for 4 weeks, that is about 28 cycles at least.
- In vitro osteogenesis with MSCs is not sufficient for the 14-day culture. This only applies to earlier-stage osteogenesis. They should go up to at least 21 days or ideally 28 days.
- In vivo it seems that CSCM-M group also had better vascularization, why? Does the magnetic field also improve vascularization? They should show relevant data/evidence supporting such observations. Can the histology images be further quantified somehow to gain additional quantitative information?
- The authors only used cranial defect to test the membrane. Is this the primary application given that the membrane would be hard to apply to any other bone defects? If so it should be made clear everywhere including the title to avoid misleading the readership. Yet then, the scope becomes very narrow limiting the broader impact of the work.
- Related, there have been numerous prior publications on the use of magnetic field and/or magnetized nanomaterials to enhance bone regeneration, including the use of magnetoelectric scaffolds for osteogenesis. How this work is truly superior than previous reports remains unclear.
- There are numerous language issues that should be corrected by the authors.

Reviewer #3 (Remarks to the Author):

Comments

This manuscript demonstrates that a core-shell $\text{CoFe}_2\text{O}_4@ \text{BaTiO}_3/\text{P}(\text{VDF-TrFE})$ membrane with high magnetoelectric coupling efficiency restores the electrical microenvironment of bone regeneration to enhance osteogenesis under co-morbidity conditions. provides a novel strategy of utilizing stimuli-responsive magnetoelectric membranes to efficiently activate osteogenesis in situ and on-demand. Materials such as non-reponsive or stimulus-responsive materials have been investigated in different studies, yet while bone growth is feasible in many settings, there is a challenge for bone regeneration in the co-morbidity condition, which hampers the speed and efficiency of osteogenesis in vivo. Here, the authors directly developed and tested the core-shell structured magnetoelectric membrane which converses magnetic force to surface potential and activates osteogenesis, even under co-morbidity. I recommend this manuscript for publication in Nature Communications before some minor corrections:

1. In the method part, how to define the time of the direct current magnetic field shifting? Why choose 12 hours as the time point?
2. As we know, nanoparticle is difficult to disperse. How to avoid core-shell nanoparticles aggregation in membrane manufacture in this study?
3. Membranes that were manufactured by spreading on a glass substrate. The thickness of the membranes will affect the polarization condition and magnetoelectric properties. Thus, how to choose and control the thickness of the membrane in this study?
4. In figure1a, there is a study about photothermal material which could also have high osteogenic efficiency. Is there any difference compared with this study? Please explain the advantages of the material in this study compared with it.
5. We know that different doses of LPS produce different biological effects. Why choose this concentration of LPS in this study for the disease model and how to determine the LPS-induced systemic inflammation successfully?
6. What meaning of SEM and immunofluorescence diagrams in Supplementary Fig. 9 supposed to illustrate? Is it biocompatibility or differentiation potential? Why does the same group of cells behave seems differently?
7. In Supplementary Fig. 10, it would be better to show the stained image for ALP and alizarin red test combined with the histogram.

Point-by-point Response

We are grateful to the reviewers for the constructive comments and suggestions on our manuscript. Below, we provide a point-by-point response of the reviewers' comments.

REVIEWER COMMENTS

Reviewer #1 (Remarks to the Author):

The work by Deng et al. presents a novel strategy to induce bone regeneration in full sized skull defects by applying a flexible magnetoelectric membrane on top of the defect and stimulating with a magnetic field. The work analyses in a rat model the effect of a pre-charged membrane alone and an even more pronounced magnetoelectric effect within a magnetic field. To verify a clinical relevance, different challenged pre-clinical model systems were compared against a “WT” setting using dexamethasone or a general inflamed setting.

The manuscript is largely well written and describes in my eyes a novel strategy and concept in sufficient detail. I have, however, some major concerns that need to be addressed prior to acceptance:

1. the authors target an osteogenic efficiency (whatever this is) and give concrete numbers in Fig. 1 in a scale. I would suggest removing these numbers and clarifying in the Figure caption what you define as “osteogenic efficiency”.

Reply: Thank you very much for taking the time to review our work and provide valuable feedback. We have carefully considered your comments and have now made appropriate revisions to our manuscript. We have removed the numbers on the vertical axis in Fig. 1a, and have defined the parameter representing osteogenic efficiency as bone volume/total volume (BV/TV) in the figure caption.

Fig. 1 Design and working mechanisms of the $\text{CoFe}_2\text{O}_4@\text{BaTiO}_3/\text{P}(\text{VDF-TrFE})$ membrane, which can enhance osteoinductivity on command upon reactivation by a magnetic field .

a, Osteogenesis efficiency of CSCM at general state (red asterisk) and under co-morbidity conditions (red four-point star), as compared with electroactive materials (blue triangles), magnetic materials (green diamonds), photothermal materials (yellow hexagons), sonodynamic materials (brown triangles) and magnetolectric materials (pink dots) under general state. Materials are classified according to the material morphology. The osteogenesis efficiency is represented by the ratio of bone volume to the total volume (BV/TV). Details and values of the aforementioned materials are listed in Table S1.

2. can you explain why the osteogenic stimulation in the “WT” is comparable to the one in the “impaired” dexamethasone group?

Reply: We thank the reviewer for the insightful comment. We compared the NC group in the regular bone defect model with the NC group in the Dex-treated bone defect model. Both these two groups exhibited poor bone regeneration. The average values of BV/TV showed that the new bone formation in the Dex-4w-NC group was slightly lower than that in the Regular-4w-NC group (Response Fig. 1).

Response Fig. 1 The average values of BV/TV in the Dex-4w-NC group and Regular-4w-NC group.

3. Why is in Figure 2 the curve of the simple membrane increasing at day 14 after I continuous decline until then?

Reply: We thank the reviewer for the insightful comment. In order to determine whether the membrane could still be activated by a magnetic field despite a decrease in surface potential upon exposure to a liquid environment for 13 days, we applied a magnetic field to the membrane for 12 hours after detecting it on the 13th day. We have indicated the application of the magnetic field with an arrow on the figure for day 13, and have added a note in the figure legend.

The surface potential began to rise again on the 14th day, indicating that the magnetic field was successful in reactivating the surface potential.

Fig. 2 Characterization of CFO@BTO core-shell particles and electrical properties of the CFO@BTO/P(VDF-TrFE) membranes.

... **h**, Zeta potential and surface potential measured by SKPM (**i**) of CSCM under continuous exposure to magnetic field, 12 h periodic magnetic field and without magnetic field respectively. On the 13th day, a magnetic field was applied to the non-magnetic field exposed group, to observe its reactivation effect on the surface potential.

4. In Figure 4 I would suggest for BV/TV similar scales to ease comparison – same applies for the other graphs in Figure 5.

Reply: Thanks for the valuable advice. We have now revised the BV/TV scales for better comparison. (Fig. 4e, Fig. 5c and Fig. 5d).

5. All groups are compared against the “WT” setting of such defects but the endogenous potential of healing in the non-treated is quite well. To allow for an estimation of the clinical potential, a control group that employs a clinical used therapy would be helpful.

Reply: We thank the reviewer for the constructive suggestions. We have now added membrane materials that are commonly used in the clinic, as comparisons with CSCMs in the rat mandibular defect model. The non-degradable membrane was represented by commercial e-PTFE, which is frequently used in bone grafting and plastic surgery. The degradable membrane was represented by the collagen membrane, which is also commonly used in the clinic. We purchased commercially-available collagen membrane from Dentium Corporation for this study. Our results showed that the external magnetic field-controlled CSCM was the most effective in promoting osteogenesis when compared to these commercially-available membrane materials (Supplementary Fig.18-19).

Supplementary Fig. 18 Comparison of osteogenic effects of CSCM versus commercially-available membranes in promoting bone defect repair. The model of mandibular critical defect (3 mm) combined with Dex-induced osteogenic inhibition in rats were used in this study. **a**, Three-dimensional micro-CT images at four weeks after material implantation, with and without magnetic field treatment (4 weeks). The bone defect areas are marked by a red dotted circle. **b**, Two-dimensional images of the defect areas at 4 weeks after material implantation and co-morbidity model preparation. The bone defect areas are marked by circles or rectangles. S=Sagittal plane, C=Coronal plane, A=Cross section. **c**, Quantitative statistics of the ratio of new bone volume to total volume (BV/TV).

Supplementary Fig. 19 Comparison of osteogenic effects of CSCM versus commercially-available membranes in promoting bone defect repair. The model of mandibular critical defect (3 mm) combined with LPS-induced systemic inflammation in rats were used in this study. **a**, Three-dimensional micro-CT images at four weeks after material implantation, with and without magnetic field treatment (4 weeks). The bone defect areas are marked by a red dotted circle. **b**, Two-dimensional images of the defect areas at 4 weeks after material implantation and co-morbidity model preparation. The bone defect areas are marked by circles or rectangles. S=Sagittal plane, C=Coronal plane, A=Cross section. **c**, Quantitative statistics of the ratio of new bone volume to total volume (BV/TV).

Reviewer #2 (Remarks to the Author):

In this manuscript, the authors presented a strategy for in situ on-demand activation of flexible magnetoelectric membrane that was shown to enhance bone defect repair. For bone defect repair under co-morbidity conditions, the use of biomaterials that can be non-invasively regulated is highly desirable to avoid further complications and to promote osteogenesis. However, there are some challenges in clinical applications to

achieve efficient osteogenesis with stimuli-responsive materials. The authors claimed to have developed polarized $\text{CoFe}_2\text{O}_4@\text{BaTiO}_3/\text{P}(\text{VDF-TrFE})$ core-shell composite membranes (CSCM) with high magnetoelectric conversion efficiency for activating bone regeneration on demand. An external magnetic field force conducted on the CoFe_2O_4 core could increase charge density on the BaTiO_3 shell and strengthens the β -phase transition in the $\text{P}(\text{VDF-TrFE})$ matrix. This energy conversion increased the membrane surface potential, which hence activated osteogenesis. *In vivo* experiments confirmed that repeated applications of external magnetic field on CSCM can enhance bone defect repair, even when osteogenesis repression is elicited by dexamethasone or lipopolysaccharide-induced inflammation.

Overall, this study provides some insights into smart biomaterials for activating osteogenesis *in vivo*, which is generally interesting to the field although there exist significant issues that shall be taken care of by the authors. At this stage, the suitability of the manuscript for the target journal is questionable. Detailed comments follow below:

1. The phrasing is inaccurate in many places. For example, they termed the membranes “core-shell composite membranes (CSCM)”, which is not correct. The particles were core-shell not the membranes. The membranes were only encapsulating these core-shell particles yet the membranes themselves were not core-shell. The language is highly misleading. In fact, how the particle parameters and membrane parameters were optimized remains elusive. How do they determine the amount of particles within the membranes? Why were the specific sizes of the core/shell chosen? How was the membrane thickness selected? None of these parameters were carefully examined by the authors leaving the readership only in perplex without understanding the rationale behind the fundamentals of the study.

Reply: Thanks for the valuable feedback from the reviewer. We have addressed the comments point-by-point in the following section:

1. Thanks for the suggested correction. We have revised the name of the membrane containing CFO@BTO core-shell nanoparticles to "core-shell particle-incorporated composite membrane (CSCM)", and the name of the membrane containing CFO particles to "CFO particle-incorporated composite membrane (CCM)". These changes were made to avoid potential confusion for readers.

2. We selected the optimal contents of particles based on previous studies (Response Fig. 2), and the amount of particles within the membrane was determined by the mass proportion of particles to matrix. The proportion of particles was optimized by testing the performance of membranes with different particle contents, magnetoelectric coupling coefficients (α) (Response Fig. 2) and mechanical properties (Supplementary Fig. 3d).

Response Fig. 2 Magnetoelectric coupling coefficients (α) of membranes with different particle mass component ratios. CSCM (10% wt) showed the highest α value.

3. The thickness of the membrane is a crucial factor that affects the magnetoelectric coupling coefficient (α)^[1]. Thicker membranes tend to have poorer magnetoelectric coupling performance under the same conditions. Conversely, if membranes are overly thin, they are easily broken-down during polarization due to weak mechanical properties. Therefore, we selected a thickness of 50 μm that was sufficiently thin, but with good mechanical properties.

4. Our parameter selection regarding particle size was based on reference to relevant research^[2, 3]. Particle size determines whether CFO can be well-coated by BTO. We tried various ratios before selecting this range of particle size, which led to a good coating and optimal magnetoelectric coupling. We thank the reviewer for the comment and will continue to update the material in future studies.

2. Similarly, they said TEM shows the cross-sectional views of the particles, which is incorrect. TEM shows projection views not necessarily a layer or surface of the cross-section.

Reply: We thank the reviewers for their valuable suggestions. As presented in the manuscript, the two TEM images contained two sets of information. The TEM images in Fig. 2a-b showed the core-shell structure, lattice conformation of the particles from a projection view and the particle morphology, while the other TEM image in Fig. 2c showed a cross-sectional view of the membrane. To obtain the latter image, the membranes were embedded, sectioned lengthwise, and observed by TEM^[4]. A schematic diagram (Response Fig. 3) is presented below to show the membrane cross-section under TEM observation. We have now revised the manuscript to avoid confusion about the two TEM images.

“...The high-resolution Transmission Electron Microscope (TEM) was used to visualize the core-shell structure and particle morphology of CSNP (Fig. 2a-b).

...Compared with CFO nanoparticle-filled composite membranes (CCM), the CSNP were uniformly distributed within the membranes, as can be observed from the surface morphology with Scanning Electron Microscopy (SEM) (Supplementary Fig. 1b-c) and the TEM view of the membrane sections (Fig. 2c).”

Response Fig. 3 The schematic diagram of the membrane cross-section under TEM

observation

3. The wording 'on-demand' is also very problematic. From the study designs, it was not used 'on-demand' but rather programmed. It seems that magnetic field was applied for 12 h then off for 12 h and cycled for 14 days or 28 days. There was nothing on-demand. On-demand means that they could have the membrane implanted, and only activates it at a certain time point oftentimes not from the beginning of implantation. For example, can they keep the membranes there for some days without activation and then activate to induce bone regeneration, and compare with the group without any activation to see how the delayed activation may 'on-demand' improve osteogenesis? Or can the membrane be pre-implanted, followed by injury, then activation of the membrane to induce osteogenesis 'on-demand'?

Reply: We thank the reviewers for their valuable suggestions. We have made the appropriate revisions by changing the term "on-demand" to "on-command", which implies that the magnetic field-modulated CSCM can effectively restore the electrical microenvironment in the bone defect area as required. The magnetic fields can be loaded at the user's command. "Programmed" is also a good suggestion, but we did not necessarily load the magnetic field every 12 hours, which is selected according to the day and night duration in this study. We can also change the cycle according to the required situation.

4. Related to the last concern, the degradation or stability profiles of these membranes remain elusive. Likely the membranes would not quite degrade *in vivo*. Yet how is their stability in terms of water uptake, or more importantly, surface adsorption of proteins/other molecules that may significantly change their charging profiles? The authors should carefully study the long-term membrane properties including activation abilities in physiological media that are rich in body fluid molecules to explain these questions and benefit the readership.

Reply: Thanks for the professional comment. The membrane material used in this study is non-degradable and requires removal after bone healing. Based on your

suggestion, we soaked the material in cell culture medium, which contained various proteins and molecules, to investigate whether surface adsorption of these molecules can affect the surface potential. As depicted in Supplementary Fig.16, the surface potential of the membranes immersed in the culture medium decreased over time. The trend of zeta potential change was consistent with that of immersion in PBS (Fig. 2h). Both membranes loaded with continuous magnetic field and 12-hour interval magnetic field were able to maintain the surface potential better than the non-magnetic field exposed group.

Supplementary Fig. 16 The surface potentials of the membranes immersed in culture medium decreased over time.

5 Are the membranes truly rechargeable? For actual experiments the authors used 12/12 h magnetic field for up to 4 weeks, but in Figure 3 they only showed a couple polarization cycles. Upon M-removal the peak was only dropping slightly still maintaining a high level and then upon M-reload it went up to the peak. How about more cycles? How many cycles can they do this? If they used 12/12 h cycles for 4 weeks, that is about 28 cycles at least.

Reply: Thank you for your comments. The term "rechargeable" is used to describe the reactivation of surface potential by an external magnetic field. This process mitigates the decline of membrane potential *in vivo* and restores the required electrical microenvironment for the bone defect area. Thanks to your suggestion, we have now

extended the experimental period to 28 days to detect the surface potential. (Supplementary Fig.16).

Regarding your comment on Figure 3, to delineate the internal mechanisms within the material, we analyzed the β -phase transition. The phenomenon of reactivation of the surface potential by the magnetic field is due to β -phase transition. We have added about 28 days of cycles to demonstrate that the magnetic field modulates the surface potential by enhancing the β -phase transition (Supplementary Fig. 17).

Supplementary Fig. 17 Relative area of β -phase calculated from XRD. The β -phase transition was detected from day 7 to day 28 after the magnetic field was removed and 3 cycles were performed.

6 *In vitro* osteogenesis with MSCs is not sufficient for the 14-day culture. This only applies to earlier-stage osteogenesis. They should go up to at least 21 days or ideally 28 days.

Reply: Thanks for your valuable comment. After cell culture for 14 days or longer on the material surface *in vitro*, the cells proliferated and covered the material, leading to mineralization. This would be detrimental to the extraction of proteins and RNA inside the cells. In accordance with the standard protocol referenced in the literature, we also studied osteogenesis within the first 14 days^[5, 6]. Promoting early osteogenic states is key to achieving the ultimate bone-forming effect. The internal responses of

cells mainly occurs within the first 14 days^[7].

We provided the results of alizarin red staining after 21 days of culture, to evaluate the level of cell mineralization on the surface of the material, as well as *in vivo* experiments on new bone formation at 4 and 8 weeks post-implantation.

7 *In vivo* it seems that CSCM-M group also had better vascularization, why? Does the magnetic field also improve vascularization? They should show relevant data/evidence supporting such observations. Can the histology images be further quantified somehow to gain additional quantitative information?

Reply: Thanks for the valuable comments. Our study primarily focused on the osteogenic properties of magnetoelectric materials. We apologize that there may be some ambiguity in our description of the H&E staining and Masson's trichrome staining. We did not mean to say that the CSCM-M group have better vascularization. Rather, we intended to describe the observation of new bone formation containing new blood vessels in the manuscript, based on our findings that new bone formation was better in the CSCM-M group. We have now revised the sentence to: "The H&E staining and Masson's trichrome staining (Supplementary Fig. 14c,d) revealed the presence of newly-formed dense bone, accompanied by bone trabeculae or vascular lumens. New bone formation was particularly conspicuous and prominent in the CSCM-M group." The information conveyed by histological staining is mainly used to evaluate the quality of new bone formation, and cannot be assumed as a vascularization-related evaluation.

8 The authors only used cranial defect to test the membrane. Is this the primary application given that the membrane would be hard to apply to any other bone defects? If so it should be made clear everywhere including the title to avoid misleading the readership. Yet then, the scope becomes very narrow limiting the broader impact of the work.

Reply: Thanks for your insightful question. To demonstrate the applications of this material, we conducted experiments on rat mandibular defects and applied DEX and

LPS for osteogenesis inhibition and systemic inflammation modeling respectively. The experimental results are presented in the aforementioned figure (Supplementary Fig. 18-19), which also demonstrates the high efficacy of CSCM-M for mandibular defect repair. We aimed to develop a material that can efficiently promote bone defect repair. The composite membrane in this study is suitable for a variety of clinical scenarios. It is trimmable with good flexibility, allowing the membrane material to fit the defect area with different shapes. Based on its trimmability, flexibility, and removability, the membranes in our study can easily be utilized for various clinical applications. Therefore, this membrane material could have a broad range of applications in guiding bone tissue regeneration.

9. Related, there have been numerous prior publications on the use of magnetic field and/or magnetized nanomaterials to enhance bone regeneration, including the use of magnetoelectric scaffolds for osteogenesis. How this work is truly superior than previous reports remains unclear.

Reply: Thanks for your valuable comments. Compared with magnetic field and/or magnetized nanomaterials, CSCM has excellent tunable electrical properties, is non-degradable, removable, and has great biocompatibility, which is advantageous for various clinical applications. Furthermore, compared with magnetoelectric scaffolds, the CSCM has superior properties conferred by membrane materials, as well as higher osteogenic efficiency, as shown in Fig 1a. We previously performed a systematic review with meta-analysis of relevant *in vivo* osteogenic effects of magnetic, electrical, and magnetoelectric materials^[8], and found that magnetoelectric membranes with magnetic field modulation had the highest efficacy.

In summary, the membrane used in this study has demonstrated high efficacy for bone regeneration, good applicability for ease of operation in clinics, and stimulus-responsive tunability to provide a conducive electric microenvironment for bone defect healing.

10 There are numerous language issues that should be corrected by the authors.

Reply: We sincerely thank the reviewer for careful reading of the manuscript text. We have now corrected the language issues in the manuscript.

Reviewer #3 (Remarks to the Author):

Comments

This manuscript demonstrates that a core-shell $\text{CoFe}_2\text{O}_4@ \text{BaTiO}_3/\text{P}(\text{VDF-TrFE})$ membrane with high magnetoelectric coupling efficiency restores the electrical microenvironment of bone regeneration to enhance osteogenesis under co-morbidity conditions. provides a novel strategy of utilizing stimuli-responsive magnetoelectric membranes to efficiently activate osteogenesis in situ and on-demand. Materials such as non-reponsive or stimulus-responsive materials have been investigated in different studies, yet while bone growth is feasible in many settings, there is a challenge for bone regeneration in the co-morbidity condition, which hampers the speed and efficiency of osteogenesis *in vivo*. Here, the authors directly developed and tested the core-shell structured magnetoelectric membrane which converses magnetic force to surface potential and activates osteogenesis, even under co-morbidity. I recommend this manuscript for publication in Nature Communications before some minor corrections:

1. In the method part, how to define the time of the direct current magnetic field shifting? Why choose 12 hours as the time point?

Reply: We thank the reviewer for the insightful question. Geomagnetic fields could change circadian rhythm and exert some biological effects^[9]. In order to simulate geomagnetic changes in the natural environment, we loaded the magnetic field for 12 hours every 24 hours.

2. As we know, nanoparticle is difficult to disperse. How to avoid core-shell nanoparticles aggregation in membrane manufacture in this study?

Reply: We thank the reviewer for the insightful question. Magnetic particles tend to

adhere and aggregate with each other easily. Therefore, we utilized the method of dispersing the suspension with an ultrasonic crusher first, followed by subjecting it to mechanical agitation under high-power ultrasonic conditions. This was done to ensure that the core-shell nanoparticles are dispersed as evenly as possible. Additionally, the presence of BTO shells on the core-shell nanoparticles can shield the magnetic attraction of the inner CFO, and the interfacial charges between BTO shells may repel each other to reduce the aggregation between particles within the membrane.

3. Membranes that were manufactured by spreading on a glass substrate. The thickness of the membranes will affect the polarization condition and magnetoelectric properties. Thus, how to choose and control the thickness of the membrane in this study?

Reply: Thanks for the critical comments. The thickness of the membranes have an impact on the magnetoelectric coupling coefficient (α)^[1]. A thicker membrane will have a lower magnetoelectric coupling coefficient, which could result in insufficient magnetoelectric coupling. Conversely, a thinner membrane may not be able to sustain the polarization treatment. Based on our previous studies, we have chosen to use a 50 μm thick membrane for this study.

4. In figure 1a, there is a study about photothermal material which could also have high osteogenic efficiency. Is there any difference compared with this study? Please explain the advantages of the material in this study compared with it.

Reply: We thank the reviewers for their insightful questions. In comparison to the photothermal material shown in Fig. 1a, the BV/TV result at 12 weeks *in vivo* was not as high as the BV/TV result at 8 weeks in our study. Apart from the shorter osteogenesis time, the magnetic field utilized in our study has higher controllability and can prevent side effects caused by overheating, such as damage to the tissue surrounding the bone defect. A detailed comparison has now been provided in the table below:

Material name	Morphology	Stimulus type	BV/TV	Time span	Ref.
GdPO ₄ /CS/Fe ₃ O ₄	Scaffold	Photothermal	61.23%	12 weeks	[10]
CFO@BTO/P(VDF-TrFE)-M	Membrane	Magnetolectric	69.33%	8 weeks	This study

5. We know that different doses of LPS produce different biological effects. Why choose this concentration of LPS in this study for the disease model and how to determine the LPS-induced systemic inflammation successfully?

Reply: We thank the reviewer for the insightful question. The LPS-induced systemic inflammation model is a well-established technique for preparing inflammatory models, with appropriate dosages and injection times being well-established in the scientific literature^[11-13]. The specific injection dosage was determined based on the body weight of the rats. On the 7th day following modeling and injection of LPS, blood was collected from the rat's caudal vein and blood routine assays were conducted. The increased white blood cell count demonstrated that the LPS-treated rats had systemic inflammation. The result is presented in Supplementary Fig. 15c as follows:

Supplementary Fig. 15c White blood cell count at 7 d after membrane implantation and LPS injection. (LPS-1 mg/kg LPS injection, NS-normal saline injection, Blank-Without any injection).

6. What meaning of SEM and immunofluorescence diagrams in Supplementary Fig. 9 supposed to illustrate? Is it biocompatibility or differentiation potential? Why does the same group of cells behave seems differently?

Reply: Thanks for the insightful question. The SEM and immunofluorescence images

were used to demonstrate good biocompatibility of the material and the ability of cells to adhere well to the material surface. Compared with 2CSCM (20% wt) and 1/2CSCM (5% wt), we found that cells on the surface of CSCM had a smaller nucleus-to-cytoplasm ratio (Response Fig. 4), which represents better osteogenic differentiation potential^[14]. The cells used for SEM imaging underwent dehydration treatment with high concentrations of sucrose and different concentrations of gradient ethanol during sample preparation, which caused some degree of cell shrinkage^[15]. The cell shrinkage led to differences in observed cell morphologies under SEM, as compared to immunofluorescence.

Response Fig. 4 Quantitative statistics of the nucleo-cytoplasmic ratio of MSC on the surfaces of the magnetoelectric composite membranes with different mass components.

7. In Supplementary Fig. 10, it would be better to show the stained image for ALP and alizarin red test combined with the histogram.

Reply: We thank the reviewer for the professional suggestion. Due to the dark color of this material, direct observation of the staining results is not possible. Therefore, the staining degree of ALP and alizarin red can only be compared through quantitative detection after staining.

Reference

- [1] Ma J, Hu J, Li Z, et al. Recent progress in multiferroic magnetoelectric composites: from bulk to thin films [J]. *Adv Mater*, 2011, 23(9): 1062-87.
- [2] Zhou D, Hao L, Gong S, et al. Magnetoelectric effect of the multilayered $\text{CoFe}_2\text{O}_4/\text{BaTiO}_3$ composites fabricated by tape casting [J]. *Journal of Materials Science: Materials in Electronics*, 2012, 23(12): 2098-103.
- [3] Selvi M M, Manimuthu P, Kumar K S, et al. Magnetodielectric properties of CoFe_2O_4 - BaTiO_3 core-shell nanocomposite [J]. *Journal of Magnetism and Magnetic Materials*, 2014, 369: 155-61.
- [4] Hamada S. PREPARATION OF TEM SPECIMEN BY CROSS-SECTION TECHNIQUE [J]. *Journal of the Atomic Energy Society of Japan*, 1986, 28(12): 1165-71.
- [5] Zhang X, Jiang W, Xie C, et al. Msx1(+) stem cells recruited by bioactive tissue engineering graft for bone regeneration [J]. *Nat Commun*, 2022, 13(1): 5211.
- [6] Gong T, Xie J, Liao J F, et al. Nanomaterials and bone regeneration [J]. *Bone Research*, 2015, 3.
- [7] Claes L, Recknagel S, Ignatius A. Fracture healing under healthy and inflammatory conditions [J]. *Nat Rev Rheumatol*, 2012, 8(3): 133-43.
- [8] Zhu F, Liu W, Li P, et al. Electric/Magnetic Intervention for Bone Regeneration: A Systematic Review and Network Meta-Analysis [J]. *Tissue engineering Part B, Reviews*, 2023.
- [9] Krylov V V. Biological effects related to geomagnetic activity and possible mechanisms [J]. *Bioelectromagnetics*, 2017, 38(7): 497-510.
- [10] Zhao P-P, Ge Y-W, Liu X-L, et al. Ordered arrangement of hydrated GdPO_4 nanorods in magnetic chitosan matrix promotes tumor photothermal therapy and bone regeneration against breast cancer bone metastases [J]. *Chemical Engineering Journal*, 2020, 381: 122694.
- [11] Mortazavi A, Mohammad Pour Kargar H, Beheshti F, et al. The effects of carvacrol on oxidative stress, inflammation, and liver function indicators in a systemic inflammation model induced by lipopolysaccharide in rats [J]. *International journal for vitamin and nutrition research Internationale Zeitschrift fur Vitamin- und Ernährungsforschung Journal international de vitaminologie et de nutrition*, 2021: 1-11.
- [12] Saadat S, Beheshti F, Askari V R, et al. Aminoguanidine affects systemic and lung inflammation induced by lipopolysaccharide in rats [J]. *Respiratory Research*, 2019, 20.
- [13] Behrends D A, Hui D, Gao C, et al. Defective Bone Repair in C57Bl6 Mice With Acute Systemic Inflammation [J]. *Clinical Orthopaedics and Related Research*, 2017, 475(3): 906-16.
- [14] Downes A, Mouras R, Elfick A. Optical spectroscopy for noninvasive monitoring of stem cell differentiation [J]. *J Biomed Biotechnol*, 2010, 2010: 101864.
- [15] Parameswaran S, Verma R S. Scanning electron microscopy preparation protocol for differentiated stem cells [J]. *Anal Biochem*, 2011, 416(2): 186-90.

REVIEWERS' COMMENTS

Reviewer #1 (Remarks to the Author):

Due to a comment from one of the reviewers, the authors changed the title and text to "on-command" instead of "on-demand".

I must admit that both versions imply that a stimulus is switched on during a biological process of regeneration and by the timing would influence the biology. To my understanding this has not been focus of this piece of work and is not illustrated/supported by data in detail. I would suggest to avoid such terminology if the data is not given in the document.

All other points that I raised have been addressed - thank you very much.

Reviewer #2 (Remarks to the Author):

The authors addressed the comments okay. In many cases there were arguments but not additional support supplies. Most might be acceptable yet one issue remaining that shall be further taken care of, as detailed below:

- In response to Q4, the authors now did stability assay over 28 days. They however, did not directly respond to the actual question – it did seem that the stability decrease was significant over the period – in the best scenario it was 50% reduction. And this was only in vitro – in vivo the degradation of the stability (not material) was perhaps much more significant in vivo. They did 8 weeks of experiments, twice long. Were they sure that the device was still useful after a few weeks in vivo? This is a compelling issue that shall be carefully examined by the authors to further clarify the true effects of their system.

Reviewer #3 (Remarks to the Author):

The authors have addressed the questions well, and I am glad to accept it at the current stage.

Point-by-point Response

We highly appreciate the reviewers' consideration and guidance on our manuscript. Below, we provide a point-by-point response of the reviewers' comments.

REVIEWERS' COMMENTS

Reviewer #1 (Remarks to the Author):

Due to a comment from one of the reviewers, the authors changed the title and text to "on-command" instead of "on-demand".

I must admit that both versions imply that a stimulus is switched on during a biological process of regeneration and by the timing would influence the biology. To my understanding this has not been focus of this piece of work and is not illustrated/supported by data in detail. I would suggest to avoid such terminology if the data is not given in the document.

All other points that I raised have been addressed - thank you very much.

Reply: Thank the reviewer for your valuable suggestions. We have carefully considered your insightful comments and have made modification on this issue. We eliminated the word "on-command" from the title and the manuscript to avoid causing undesired misleading. Thank you again for your professional guidance, which makes our manuscript more rigorous and correct.

Reviewer #2 (Remarks to the Author):

The authors addressed the comments okay. In many cases there were arguments but not additional support supplies. Most might be acceptable yet one issue remaining that shall be further taken care of, as detailed below:

- In response to Q4, the authors now did stability assay over 28 days. They however,

did not directly respond to the actual question – it did seem that the stability decrease was significant over the period – in the best scenario it was 50% reduction. And this was only in vitro – in vivo the degradation of the stability (not material) was perhaps much more significant in vivo. They did 8 weeks of experiments, twice long. Were they sure that the device was still useful after a few weeks in vivo? This is a compelling issue that shall be carefully examined by the authors to further clarify the true effects of their system.

Reply: Thanks for the professional comments. In response to your valuable suggestion in Question 3 of the initial review, we conducted an extensive 28-day monitoring of the zeta surface potential. Using the experimental data obtained, we performed exponential two-phase decay curve fitting. Subsequently, we utilized statistical analysis to predict the surface potential for the upcoming four weeks. (Response Fig. 1). The results showed that, even by the 8th week, both membranes loaded with continuous magnetic field and 12-hour interval magnetic field could maintain the surface potential within the range of physiological potential required for bone reconstruction.¹ This trend is consistent with the fact that a higher surface potential is required in the early stage of bone defect repair, and mineralization and bone maturation tend to be stable in the later stage². The predicted 8-week surface potential can also meet the potential requirements for bone reconstruction.¹

Additionally, it is worth noting that there was a significant statistical difference between the CSCM-M group and the NC group at 4 weeks ($p < 0.001$). Similarly, at 8 weeks, there was a significant statistical difference between the CSCM-M group and the NC group ($p < 0.0001$). These findings suggest that the material remained effective even after 4 weeks.

Response Fig. 1 The fitting and simulating of zeta surface potential in the following 4 weeks. The pink area represents the experimental detection potential data of previous four weeks, while the blue area represents the predicted value after fitting an exponential two-phase decay curve ($R^2=0.9434$ in the group Continuous magnetic field, $R^2=0.9568$ in the group 12 h periodic magnetic field, $R^2=0.9390$ in the group Absence of magnetic field).

Reviewer #3 (Remarks to the Author):

The authors have addressed the questions well, and I am glad to accept it at the current stage.

Reply: Thank you very much for your comments and professional advice. These opinions help us to improve academic rigor of our manuscript.

Reference

- 1 Akshata, C. R., Harichandran, G. & Murugan, E. Effect of pectin on the crystallization of strontium substituted HA for bone reconstruction application. *Colloids Surf B Biointerfaces* **226**, 113312 (2023). <https://doi.org/10.1016/j.colsurfb.2023.113312>
- 2 Jia, F. *et al.* Comprehensive Evaluation of Surface Potential Characteristics on Mesenchymal Stem Cells' Osteogenic Differentiation. *ACS Appl Mater Interfaces* **11**, 22218-22227 (2019). <https://doi.org/10.1021/acsami.9b07161>